# Mean Flow Distillation:
# Robust and Stable Distillation for Flow Matching Models

**An Zhao** [1]   **Shengyuan Zhang** [1]   **Zhongjian Sun** [2]   **Yixiang Zhou** [1]   **Zejian Li** [2]   **Ling Yang** [3]   **Tianrun Chen** [1]
**Lingyun Sun** [4]

## Abstract

Flow Matching models have demonstrated strong performance across a wide range of generative tasks. However, their reliance on ODE-based iterative sampling incurs substantial computational overhead in inference, which limits their applicability in real-time scenes. While distillation is a promising solution, existing approaches largely borrow from diffusion-based score matching, often failing to exploit the intrinsic geometric structure of flows and suffering from training instability, high variance, and degraded generation quality. In this paper, we propose Mean Flow Distillation (MFD), a novel distillation framework tailored for flow matching models. We theoretically demonstrate that MFD acts as a temporal low-pass filter, effectively suppressing the high-frequency optimization noise inherent in variational score distillation (VSD) while ensuring global trajectory consistency. We further prove the Mean Flow Matching Theorem, establishing that matching expected average velocities is sufficient for strict distribution alignment. Empirically, on challenging tasks of high-dimensional manifolds including 4D occupancy forecasting and text-to-image generation, MFD achieves state-of-the-art performance, enabling high-fidelity single-step generation. The implementation is available at https://github.com/happyw1nd/MFD.

## 1. Introduction

In recent years, flow-based generative modeling based on Flow Matching (FM) and stochastic interpolants (Liu et al., 2023; Lipman et al., 2023; Albergo & Vanden-Eijnden, 2023) has emerged as a powerful paradigm, demonstrating remarkable performance in applications like image generation (Qin et al., 2025b) and 3D modelling (Xiang et al., 2025). Unlike score-based diffusion models (Yin et al., 2024a; Wang et al., 2023), FM directly learns a vector field connecting marginal distributions (Liu et al., 2023; Lipman et al., 2023), offering a more explicit dynamical description and superior generation quality. However, FM relies on time-consuming iterative sampling (Luo et al., 2023; Lipman et al., 2024). As model scales grow, high inference latency limits FM's applicability in real-time or resource-constrained scenarios (Zhou et al., 2025b).

The existing acceleration methods, such as advanced schedulers (Liu et al., 2022; Lu et al., 2022) or truncation (Zheng et al., 2023), often fail to maintain quality in few-step regimes. Distillation remains the most promising strategy (Yin et al., 2024a; Luo et al., 2023), yet existing approaches are primarily designed for score-based formulations. Directly adapting these to FM often involves indirect score conversions (Zhou et al., 2025b), failing to exploit FM's intrinsic geometric structures. This mismatch typically leads to training instability, high variance, and degraded generation quality (Ge et al., 2026).

To address these challenges, we propose **Mean Flow Distillation (MFD)**, a general-purpose framework tailored for pretrained FM models. Unlike methods that forcibly match instantaneous velocity fields or rely on high-variance score estimates (Ge et al., 2026; Yang et al., 2025), MFD utilizes Mean Flow, the time-integrated velocity field, as the core alignment metric. Our key insight is that the cumulative displacement induced by the mean flow provides a stable, global consistency constraint, effectively filtering out local geometric perturbations common in instantaneous matching.

Technically, MFD alternates between a learnable Auxiliary Flow Model, which approximates the student-induced vector field, and the student, which matches the auxiliary

---
[1]College of Computer Science and Technology, Zhejiang University, Zhejiang, China [2]School of Software Technology, Zhejiang University, Zhejiang, China [3]Princeton University, Princeton, USA [4]College of Artificial Intelligence, Zhejiang University, Zhejiang, China. Correspondence to: Zejian Li <zejianlee@zju.edu.cn>.

*Proceedings of the 43rd International Conference on Machine Learning*, Seoul, South Korea. PMLR 306, 2026. Copyright 2026 by the author(s).

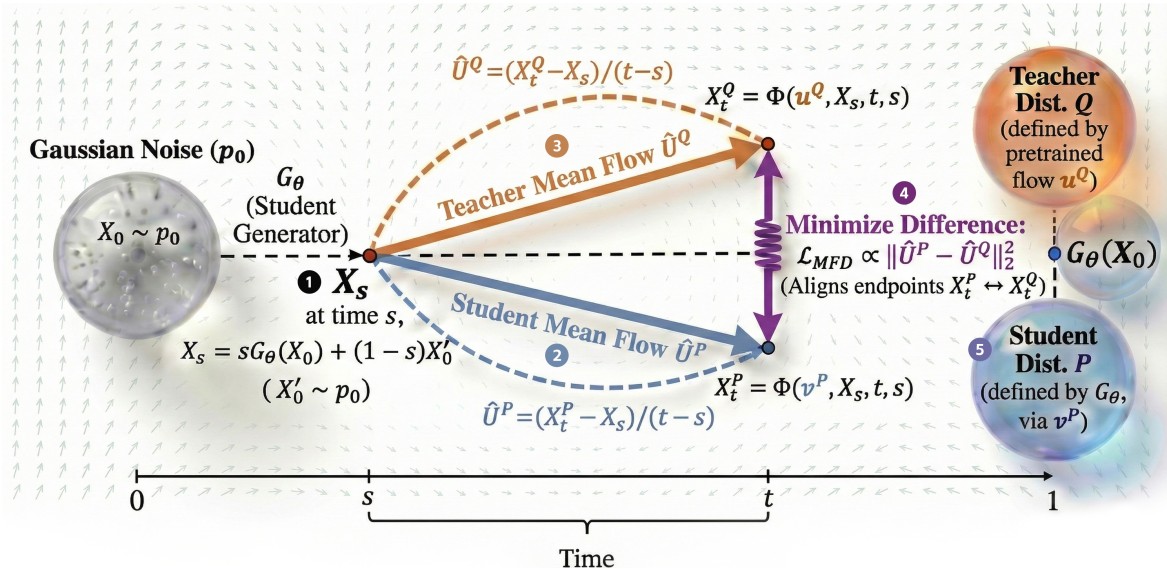

*Figure 1.* Overview of Mean Flow Distillation (MFD). **(1)** Sample noise $X_0 \sim \mathcal{N}(0, I)$, generate $X_1$ via the single-step student model $G_\theta$, and construct intermediate state $X_s$ through linear interpolation following Eq (2) with $t$ replaced by $s$. **(2)** Compute the auxiliary mean flow by integrating the auxiliary model $v^P$ from $X_s$ over interval $[s, t]$ via ODE solver $\Phi$ following Eq (16). **(3)** Compute the teacher mean flow by integrating the pretrained teacher model $u^Q$ from the same $X_s$ over $[s, t]$ also following Eq (16). **(4)** Optimize the student model $G_\theta$ by matching the two mean flows, with gradients backpropagated through $X_s$ following Eq (20). **(5)** Update the auxiliary model $v^P$ to track the evolving student distribution $P$ following Eq (3). This temporal integration over $[s, t]$ provides more robust and low-variance gradients compared to instantaneous velocity matching, enabling stable distillation with high-quality generation.

model's average velocity to that of a frozen teacher via multi-step ODE integration. The auxiliary model is used only during training and can be implemented with lightweight adapters in large-scale settings (Section 5.3). Theoretically, MFD acts as a temporal low-pass filter that suppresses high-frequency optimization noise. We establish a Mean Flow Matching Theorem, proving that matching the expected average velocity is sufficient for strict distribution alignment, and show that MFD yields lower variance than variational score distillation (VSD) (Wang et al., 2023).

Our contributions are summarized as follows:

- We propose Mean Flow Distillation (MFD), the first distillation paradigm that directly aligns velocity fields for Flow Matching models, avoiding indirect score conversion. By using the time-integrated mean flow as the supervision signal, MFD enables stable and high-quality distillation.

- We formalize Instantaneous Velocity Field Matching and prove the Mean Flow Matching Theorem, showing that matching the mean flow of models will achieve strict distributional alignment. We further theoretically analyze the gradient variance and demonstrate that MFD yields lower variance and more stable training dynamics than variational score distillation.

- On two challenging tasks, text-to-image generation and 4D occupancy forecasting, MFD outperforms existing distillation methods, including score-based and consistency-based flow distillation, in generation quality and distributional fidelity.

**Conflict of Interest Disclosure.** The authors declare no conflicts of interest.

## 2. Related Works

### 2.1. Flow Matching

Lipman et al. (2023) introduced Flow Matching (FM) to train continuous normalizing flows by regressing velocity fields along probability paths, while stochastic interpolants (Albergo & Vanden-Eijnden, 2023) independently developed a closely related continuous-time construction based on interpolating paths. Concurrently, Rectified Flow (Liu et al., 2023) emphasized straightening transport trajectories to approach geodesics, significantly reducing integration error for few-step sampling. FM has since matured into a general-purpose paradigm applied to text-to-image (Qin et al., 2025a; Xie et al., 2025), 3D generation (Xiang et al., 2025; Chen et al., 2025) and 4D occupancy forecasting (Liu et al., 2025).

Recent research targets one-step generation within this framework. Mean Flow (MF) (Geng et al., 2025a) replaces instantaneous velocity supervision with time-integrated mean velocity, achieving one-step generation without distillation. Improved Mean Flow (Geng et al., 2025b) further refines this by addressing target-network dependencies. Inductive Moment Matching (Zhou et al., 2025a) proposes a single-stage training procedure for few-step generation, guaranteeing distribution-level convergence without requiring pre-trained initialization. However, these methods are designed for end-to-end training from scratch. Their objectives cannot be directly applied to distill existing high-quality pretrained models, limiting their utility as general-purpose acceleration mechanisms. We provide a detailed comparison between MFD, MF, and related flow distillation methods in Section B.1.

## 2.2. Sampling Acceleration via Distillation

Distillation methods aim to compress the computationally expensive iterative sampling process into a few steps or even a single step. These approaches generally fall into two main categories: Consistency Distillation and Score Distillation.

The principle of consistency distillation is to enforce trajectory self-consistency, ensuring that points along the probability flow ODE consistently map to corresponding initial states. Progressive Distillation (Salimans & Ho, 2022) achieves acceleration by iteratively halving the number of sampling steps. Consistency Models (Song et al., 2023) propose learning a mapping from any timestep to the data origin. Latent Consistency Models (LCM) (Luo et al., 2024) apply this paradigm to latent space, while Multistep Consistency Models (MCM) (Heek et al., 2024) introduce a generalized formulation that bridges the gap between single-step and multi-step generation regimes. Align Your Flow (AYF) (Sabour et al., 2025) introduces continuous-time flow map distillation, generalizing consistency models to connect arbitrary noise while maintaining effectiveness across all steps. AYF enforces sample-level flow-map consistency, whereas MFD matches distribution-level expected average velocity fields; this distinction is analyzed in Section B.1.

Instead of relying on trajectory-level constraints, score distillation optimizes the student generator by aligning its gradients with the density fields implied by the teacher model. Variational Score Distillation (VSD) (Wang et al., 2023) utilizes the teacher as a score-based prior to guide the student's learning. Techniques such as Diff-Instruct (Luo et al., 2023), DMD (Yin et al., 2024b), and DMD2 (Yin et al., 2024a) explicitly minimize the KL divergence between the student and teacher distributions through score matching objectives. More recent advances (Sauer et al., 2024) further enhance generation quality by combining score distillation with adversarial training objectives. Additionally, Score

Identity Distillation (SiD) (Zhou et al., 2024) accelerates the distillation process by leveraging novel score identities that eliminate the need for real data components. Swift-Brush (Nguyen & Tran, 2024) demonstrates the feasibility of image-free distillation using variational score distillation.

Recently, several works apply these techniques to flow matching models (Liu et al., 2024; Ge et al., 2026). However, these methods typically necessitate converting velocity fields into score functions (Zhou et al., 2025b). This conversion is often suboptimal, as it fails to fully exploit the deterministic ODE structures intrinsic to flow matching.

## 3. Preliminaries

### 3.1. Flow Matching

Flow Matching is a class of methods that perform distribution transformation by learning ordinary differential equations (ODEs). The objective is to directly learn a velocity field that maps an initial distribution $\pi_0$ to a target distribution $\pi_1$, without relying on stochastic diffusion processes. Rectified Flow provides a simple yet effective instantiation of flow matching by directly regressing toward straight-line interpolation paths between noise and data, yielding a straightforward MSE training objective.

Consider two probability distributions $\pi_0$ and $\pi_1$ in $\mathbb{R}^d$. Flow matching tries to learn an ODE that defines a deterministic transport process mapping samples from $\pi_0$ to $\pi_1$

$$\frac{dZ_t}{dt} = v(Z_t, t), \quad t \in [0, 1], \quad (1)$$

The velocity field $v : \mathbb{R}^d \times [0, 1] \to \mathbb{R}^d$ is learned in a data-driven manner and satisfies $Z_0 \sim \pi_0$ and $Z_1 \sim \pi_1$.

The core idea of flow matching is to adopt a simple interpolation process as a reference trajectory and train a model to learn the corresponding dynamics to match this trajectory. Specifically, a linear interpolation between $X_0$ and $X_1$ is:

$$X_t = (1 - t)X_0 + tX_1, \quad (2)$$

The time derivative is constant, given by $\dot{X}_t = X_1 - X_0$. Rather than directly using this non-causal path, which depends on future information, rectified flow learns a velocity field $v(X_t, t)$ that approximates the instantaneous velocity of the interpolation trajectory by minimizing a mean squared error (MSE) objective. Concretely, the velocity field is learned by minimizing the following least-squares objective:

$$\min_v \mathbb{E}_{X_0, X_1, t \sim \mathcal{U}[0,1]} \left[ \| (X_1 - X_0) - v(X_t, t) \|^2 \right] \quad (3)$$

The optimal solution to Equation (3) admits a closed-form expression in terms of a conditional expectation:

$$v^*(x, t) = \mathbb{E}[X_1 - X_0 \mid X_t = x], \quad (4)$$

That is, conditioned on a given position $X$ and time $t$, the velocity field equals the expected direction of all linear paths passing through that point. The resulting velocity field depends only on the current state and time, thereby defining a well-posed ODE that can be simulated in a forward manner.

The trajectory $Z_t$ generated under this velocity field enjoys an important marginal preservation property: for any $t \in [0, 1]$, the random variable $Z_t$ follows the same distribution as the interpolation $X_t$. Thus, the terminal state satisfies $Z_1 \sim \pi_1$. Moreover, due to the uniqueness of ODE solutions, different trajectories do not intersect during evolution. This non-crossing property consolidates potentially entangled linear interpolation paths into a coherent deterministic flow, giving rise to the rectified flow (Liu et al., 2023).

### 3.2. Score Distillation

Score distillation is inspired by VSD (Wang et al., 2023), which aims to directly distill a multi-step teacher diffusion model into a few-step or single-step student model. Specifically, let $q$ be the ground truth distribution approximated by the pre-trained teacher model $u^Q$, $p_1$ be the generative distribution of the student model $G_\theta$, where $\theta$ is the trainable parameter. Score distillation minimizes the KL divergence between $q$ and $p_1$ (Wang et al., 2023; Yin et al., 2024a)

$$\min_\theta D_{KL} \left( p_1 \left( X_1 \right) \| q \left( X_1 \right) \right) \tag{5}$$

Here $X_1$ is the sample generated by the student model $G_\theta$ with few- or single-step generation. However, directly solving the problem in Equation (5) is difficult. According to Theorem 1 in (Wang et al., 2023), the optimization objective in Equation (5) can be expanded to different timesteps $t$

$$\min_\theta D_{KL} \left( p_t \left( X_t \right) \| q_t \left( X_t \right) \right) \tag{6}$$

Here $t \in [0, 1]$ is the noise level, and $X_t$ is the noisy version of the generated $X_1$. Thus, by expanding the KL divergence, the gradient of the student model is approximated as

$$\nabla_\theta D_{KL} \left( p_t \left( X_t \right) \| q_t \left( X_t \right) \right)$$
$$= \mathbb{E}_{t,\epsilon} \left[ \nabla_{X_t} \log p_t \left( X_t \right) - \nabla_{X_t} \log q_t \left( X_t \right) \right] \frac{\partial X_t}{\partial \theta}, \tag{7}$$

here $\epsilon \sim \mathcal{N}[0, I]$ is the random noise. Because the scores cannot be obtained directly, the teacher score $\nabla_{X_t} \log q_t \left( X_t \right)$ is approximated by a pre-trained diffusion model $s_\theta$ and the student score $\nabla_{X_t} \log p_t \left( X_t \right)$ by an auxiliary diffusion model $s_\phi$. Then, with the simplification in (Wang et al., 2023; Ho et al., 2020), the gradient of the student model is estimated by

$$\nabla_\theta D_{KL} \left( p_t \left( X_t \right) \| q_t \left( X_t \right) \right)$$
$$\approx \mathbb{E}_{t,\epsilon} \left[ \| s_\phi \left( X_t, t \right) - s_\theta \left( X_t, t \right) \| \frac{\partial X_t}{\partial \theta} \right] \tag{8}$$

The pre-trained diffusion model $s_\theta$ is the teacher model, and the auxiliary diffusion model $s_\phi$ is independently trained with the denoising loss on generated samples $X_1$ (Song et al., 2021)

$$\min_\phi \mathbb{E}_{t,\epsilon} \left\| s_\phi \left( X_t, t \right) - \frac{X_1 - X_t}{\sigma_t^2} \right\|_2^2 \tag{9}$$

During training, the student model and the auxiliary model are optimized alternately, with the optimization schedule configurable according to the specific task requirements.

Recent works apply score distillation to flow matching models (Liu et al., 2024; Ge et al., 2026). Specifically, they convert the velocity field $v(X_t, t)$ to a score function via $s(X_t, t) \approx \frac{X_t - v(X_t, t)}{(1-t)^2}$ (Liu et al., 2024). Such conversion is suboptimal for two reasons. First, division by $(1-t)^2$ amplifies velocity-estimation errors as $t$ approaches 1, directly increasing gradient variance. Second, the converted score is matched at independently sampled timesteps, discarding the deterministic ODE trajectory structure that FM models explicitly learn. MFD therefore operates directly in velocity space and integrates supervision along flow trajectories.

## 4. Mean Flow Distillation

Existing distillation methods for flow matching models typically convert the velocity field into an equivalent score function and then apply score-based methods developed for diffusion models (Zhou et al., 2025b; Ge et al., 2026). While this inherits score matching, it overlooks the intrinsic structure of flow matching (the deterministic ODE trajectories and explicit velocity field formulation). In flow matching, a velocity field $v(x, t)$ uniquely defines a deterministic transport path from a source distribution to a target distribution, implying that **two flows with identical velocity fields generate identical trajectories and distributions**. This observation motivates an intuitive distillation paradigm that directly aligns velocity fields, rather than relying on an indirect score transformation.

We formalize this idea via Instantaneous Velocity Field Matching theorem (Theorem 4.1), which provides a theoretical basis for direct velocity-based distillation. However, instantaneous matching uses supervision from a single sampled timestep. Empirically, VSD-style instantaneous baselines show wider loss distributions and more oscillatory training curves (Figure 2 and Section D.5); theoretically, their gradient variance is tied to the noise of a single instantaneous estimate (Section C.3). To address this, inspired by mean flow theory (Geng et al., 2025a), we propose Mean Flow Distillation (Theorem 4.3), which replaces pointwise velocity alignment with matching time-integrated velocity fields, yielding stable and effective distillation.

## 4.1. Problem Setup

Let $p_0$ be the source distribution, set as $\mathcal{N}(0, I)$ by default in a space $\mathbb{R}^d$, and $Q$ be the target distribution of the training samples. We consider three key entities:

The **teacher model** is a pretrained flow matching model $u^Q : \mathbb{R}^d \times [0, 1] \to \mathbb{R}^d$, which defines a probability flow transporting samples from $p_0$ to $Q$. The corresponding flow map is denoted as $\psi_t^Q$, specified by the time-dependent instantaneous velocity field of the pretrained distribution $Q$.

The **student model** $G_\theta : \mathbb{R}^d \to \mathbb{R}^d$ is a one-step generator with trainable parameters denoted as $\theta$. $G_\theta$ maps a latent noise $X_0 \sim p_0$ to a generated sample $X_1 = G_\theta(X_0)$. The marginal distribution of generated samples is defined as $P$. Both $P$ and $Q$ are defined on $\mathbb{R}^d$ and share the same source $p_0$. Intuitively, the training objective is to optimize $\theta$ such that $P$ closely matches the target $Q$.

Since the student model is a single-step mapping and does not explicitly provide probability flow information, an **auxiliary flow model** $v^P : \mathbb{R}^d \times [0, 1] \to \mathbb{R}^d$ is introduced to approximate the student distribution $P$. Without loss of generality, we assume that $v^P$ can exactly reconstruct the instantaneous velocity field of the distribution $P$, with the corresponding flow map $\psi_t^P$ satisfying $\psi_1^P(X_0) \sim P$. The use of an auxiliary flow model is both necessary and theoretically justified; see Section C.4 for the proof and Section 5.3 for its capacity and overhead in practice. The overview of Mean Flow Distillation is shown in Figure 1.

## 4.2. Instantaneous Velocity Field Matching

Building on the trajectory uniqueness property of flow matching, we first establish a theoretical foundation for flow-based distillation through instantaneous velocity field alignment. Theorem 4.1 shows that matching velocity fields pointwise is sufficient to ensure distribution matching.

**Theorem 4.1** (Instantaneous Velocity Field Matching). *Let $u^Q$ and $v^P$ be the instantaneous velocity fields corresponding to the teacher distribution $Q$ and the student distribution $P$, respectively. Assume both velocity fields satisfy Lipschitz continuity. Let $D(\cdot, \cdot)$ denote a Bregman divergence on $\mathbb{R}^d$ satisfying strict positive definiteness $(D(x, y) = 0 \iff x = y)$. If the expected Bregman divergence between the instantaneous velocity fields is zero:*

$$\mathcal{L}_{IVF} = \mathbb{E}_{t \sim \mathcal{U}[0,1], X_t \sim p_t^P} \left[ D\left(v^P(X_t, t), u^Q(X_t, t)\right) \right] = 0. \tag{10}$$

*Then the student distribution coincides with the teacher distribution, i.e., $P = Q$.*

The proof of Theorem 4.1 is shown in Section C.1.

While instantaneous velocity matching provides a theoretically sound foundation for distillation, it can be further improved in practice. Due to its reliance on pointwise velocity alignment at individual time steps, this approach exhibits sensitivity to local perturbations in $X_t$ and does not explicitly leverage the cumulative trajectory information over time intervals. These observations motivate us to develop a more robust alignment criterion that better captures the global structure of flows through temporal integration. As in Section C.3, when reduced to single-step integration, our proposed method (MFD) degenerates to Instantaneous Velocity Field Matching with comparable variance and training stability to VSD, confirming that the benefits arise specifically from multi-step temporal averaging.

## 4.3. Mean Velocity Field Matching

Recent work on Mean Flows for one-step generative modeling (Geng et al., 2025a) demonstrates that supervising with time-integrated mean velocities rather than instantaneous velocities leads to better generation quality. We apply this idea to the distillation setting with a key insight: temporal integration acts as a low-pass filter, effectively averaging out high-frequency noise in instantaneous velocity estimates and providing more robust, low-variance gradients for student model optimization.

Compared to pointwise instantaneous matching, matching time-integrated mean velocities offers stronger supervision signals that naturally capture global trajectory consistency rather than volatile local tangent alignment. While one intuitive perspective suggests that velocity errors may accumulate during integration over $[s, t]$ to amplify supervision signals, we discuss this viewpoint and its relationship to the variance reduction mechanism in Section B.5.

Consider the displacement of a particle along a flow trajectory from time $s$ to time $t$:

$$\psi_t(X_0) - \psi_s(X_0) = \int_s^t v(\psi_\tau(X_0), \tau) \, d\tau. \tag{11}$$

This displacement is determined by the cumulative effect of the velocity field over the interval $[s, t]$. We define Average Velocity Field (AVF) to capture this cumulation:

**Definition 4.2** (Average Velocity Field, AVF). *For a velocity field $v$ with flow map $\psi_t$, the average velocity field $U(z_s, s, t)$ over the interval $[s, t]$ starting from position $z_s = \psi_s(X_0)$ is defined as:*

$$U(z_s, s, t) = \frac{1}{t - s} \int_s^t v(\psi_\tau(X_0), \tau) \, d\tau$$
$$= \frac{\psi_t(X_0) - \psi_s(X_0)}{t - s}. \tag{12}$$

Intuitively, $U(z_s, s, t)$ represents the mean velocity required to transport a particle from $z_s$ to $z_t = \psi_t(X_0)$ over $[s, t]$.

AVF acts as a temporal low-pass filter on the instantaneous velocity field (detailed analysis in Section B.3). By integrating over time, it filters out high-frequency stochastic noise and optimization artifacts inherent in gradient estimation (e.g., mini-batch sampling noise, numerical instabilities) and this provides a smoother supervision signal. This motivates our main theoretical result, which shows that matching average velocity fields is sufficient for distribution alignment.

Let $U^Q(z, s, t)$ and $U^P(z, s, t)$ be the average velocity fields corresponding to the distributions $Q$ and $P$, respectively. Taking $U^P$ as an example, it satisfies the following integral relationship for $0 \leq s < t \leq 1$:

$$(t - s)U^P(z_s, s, t) = \int_s^t v^P \left( \psi_\tau^P \left( (\psi_s^P)^{-1}(z_s) \right), \tau \right) d\tau. \tag{13}$$

Intuitively, AVF describes the mean displacement vector of a particle moving from position $z_s$ to $z_t$ over the time interval $[s, t]$, i.e., $z_t = z_s + (t - s)U^P(z_s, s, t)$.

**Theorem 4.3** (Mean Flow Matching). *Let $U^Q(\cdot, s, t)$ and $U^P(\cdot, s, t)$ be the average velocity fields of $Q$ and $P$ from time $s$ to $t$. Let $D(\cdot, \cdot)$ denote a Bregman divergence on $\mathbb{R}^d$ satisfying strict positive definiteness ($D(x, y) = 0 \iff x = y$). If the expected Bregman divergence between the AVF of the auxiliary flow $P$ and the teacher flow $Q$ is zero:*

$$\mathcal{L}_{MFD} = \mathbb{E}_{s,t \sim p(s,t), X_s \sim p_s^P} \left[ D\left( U^P(X_s, s, t), U^Q(X_s, s, t) \right) \right] = 0, \tag{14}$$

*then the student distribution coincides with the teacher distribution, i.e., $P = Q$.*

Here, $p(s, t)$ denotes a distribution with full support on the time domain, and the velocity fields are assumed to satisfy Lipschitz continuity. The proof of Theorem 4.3 is provided in Section C.2. As the interval $[s, t]$ shrinks to a single point ($s \to t$), the average velocity field converges to the instantaneous. Thus, matching AVFs over intervals with full temporal support implies matching instantaneous velocities everywhere, which by Theorem 4.1 guarantees $P = Q$.

Compared to instantaneous velocity matching, Mean Flow Matching offers significant practical advantages. (1) The temporal integration averages out noise in instantaneous velocity estimates, reducing gradient variance (Section C.3). (2) AVF captures cumulative trajectory behavior for global consistency. We visualize MFD's smooth velocity field in a controlled 2D toy example (Section D.2).

### 4.4. Practical Implementation

In practice, the $\ell_2$ norm is used as the Bregman divergence. Since both the auxiliary model $v^P$ and the teacher model $u^Q$ are instantaneous velocity fields, the AVF must be approximated by numerical integration over the interval $[s, t]$.

Let $\Phi(\cdot)$ denote a numerical ODE solver. For any instantaneous velocity field $v$ (either $v^P$ or $u^Q$), the estimated average velocity $\hat{U}$ over $[s, t]$ is computed as follows:

First, starting from state $X_s$ at time $s$, we integrate forward to time $t$ to obtain the reference point $X'_t$:

$$X'_t = \Phi(v, X_s, t, s) \approx X_s + \int_s^t v(X_\tau, \tau) d\tau. \tag{15}$$

Then, the average velocity vector is computed based on the resulting displacement:

$$\hat{U}(X_s, s, t; v) = \frac{X'_t - X_s}{t - s}. \tag{16}$$

Thus, we obtain the auxiliary model's average velocity $\hat{U}^P = \hat{U}(X_s, s, t; v^P)$ and the teacher model's average velocity $\hat{U}^Q = \hat{U}(X_s, s, t; u^Q)$.

The goal of MFD is to optimize the student model $G_\theta$ such that the average velocity estimated by the auxiliary model approximates the teacher's average velocity:

$$\mathcal{L}_{\text{MFD}}(\theta) = \frac{1}{2} \mathbb{E}_{s,t,X_0}$$
$$\left[ \left\| \hat{U}^P(X_s(\theta), s, t; v^P) - \hat{U}^Q(X_s(\theta), s, t; u^Q) \right\|_2^2 \right], \tag{17}$$

where $X_s(\theta) = s \cdot G_\theta(X_0) + (1 - s) \cdot X'_0$ and $X'_0 \sim p_0$.

Let $\Delta(X_s) = \hat{U}^P(X_s(\theta), s, t; v^P) - \hat{U}^Q(X_s(\theta), s, t; u^Q)$ be the difference between the two estimated mean velocities. According to the chain rule, the gradient of the student model is:

$$\nabla_\theta \mathcal{L}_{\text{MFD}} = \mathbb{E} \left[ (\nabla_\theta X_s)^\top \cdot \nabla_{X_s} \Delta(X_s) \cdot \Delta(X_s) \right] \tag{18}$$

The term $\nabla_{X_s} \Delta(X_s)$ involves differentiating the output of the ODE solver $\Phi$ with respect to the input $X_s$. This requires backpropagation through the solver, which is computationally expensive and potentially unstable. Following a simplification similar to Score Distillation Sampling (SDS) (Poole et al., 2023), we treat the estimated average velocity fields $\hat{U}^P$ and $\hat{U}^Q$ as fixed guidance targets at the current state $X_s$. By omitting the Jacobian term (namely approximating $\nabla_{X_s} \Delta(X_s) \approx I$ effectively for the direction), the gradient in Equation (18) is simplified to:

$$\nabla_\theta \mathcal{L}_{\text{MFD}} \approx \mathbb{E} \left[ (\nabla_\theta X_s)^\top \cdot \Delta(X_s) \right] \tag{19}$$

Substituting $\nabla_\theta X_s = s \cdot \nabla_\theta G_\theta(X_0)$ into Equation (19), the final gradient for the student model is:

$$\nabla_\theta \mathcal{L}_{\text{MFD}} \approx \mathbb{E}_{s,t,X_0} \left[ s \cdot (\nabla_\theta G_\theta(X_0))^\top \cdot \right.$$
$$\left. \left( \underbrace{\frac{\Phi(v^P, X_s, t, s) - X_s}{t - s}}_{\text{Auxiliary Mean Flow}} - \underbrace{\frac{\Phi(u^Q, X_s, t, s) - X_s}{t - s}}_{\text{Teacher Mean Flow}} \right) \right] \tag{20}$$

**Algorithm 1** Mean Flow Distillation (MFD)

---

**Input:** Pretrained teacher $u^Q$, student $G_\theta$, auxiliary flow model $v_\phi$, ODE solver $\Phi$, iterations $N$, TTUR ratio $K = 10$
**Output:** Trained student model $G_\theta$
**for** $n = 1$ **to** $N$ **do**
    *// Update auxiliary flow model (K steps)*
    **for** $k = 1$ **to** $K$ **do**
        Sample $X_0 \sim \mathcal{N}(0, I)$
        **with** no grad:
            $X_1 = G_\theta(X_0)$     *// Student sample*
        Resample $X_0 \sim \mathcal{N}(0, I)$; Sample $\tau \sim \mathcal{U}[0, 1]$
        $X_\tau = (1 - \tau)X_0 + \tau X_1$
        Update $\phi$: $\mathcal{L}_{FM} = \|v_\phi(X_\tau, \tau) - (X_1 - X_0)\|_2^2$
    **end for**
    *// Update student model (1 step)*
    Sample $X_0 \sim \mathcal{N}(0, I)$; $X_1 = G_\theta(X_0)$
    Resample $X_0 \sim \mathcal{N}(0, I)$
    Sample $s, t \sim p(s, t)$ with $0 \le s < t \le 1$
    $X_s = (1 - s)X_0 + sX_1$
    **with** no grad:
        $\hat{U}^Q = (\Phi(u^Q, X_s, t, s) - X_s)/(t - s)$
        $\hat{U}^P = (\Phi(v_\phi, X_s, t, s) - X_s)/(t - s)$
    Update $\theta$: $\mathcal{L}_{MFD} = (\hat{U}^P - \hat{U}^Q) \cdot X_s$
**end for**
**return** $G_\theta$

---

*Table 1.* Comparison of MFD against the teacher model and baseline distillation methods on 4D occupancy forecasting. FPS is calculated on a single RTX 3090.

| Model | NFE ↓ | IoU ↑ (%) | mIoU ↑ (%) | FPS ↑ |
|---|---|---|---|---|
| OccFM (Teacher) | 10 | 37.52 | 28.49 | 12.72 |
| OccFM (Teacher) | 5 | 36.05 | 25.18 | 16.75 |
| OccFM (Teacher) | 2 | 29.02 | 13.72 | 21.33 |
| OccFM (Teacher) | 1 | 27.42 | 8.15 | 24.92 |
| Diff-Instruct | 1 | 36.41 | 26.42 | **25.21** |
| CD | 1 | 31.69 | 20.70 | 25.08 |
| AYF (Sabour et al., 2025) | 1 | 32.38 | 21.69 | – |
| DMD2 | 1 | 35.84 | 24.27 | 25.11 |
| SenseFlow | 1 | 35.40 | 23.91 | 25.09 |
| **MFD (Ours)** | 1 | **37.07** | **27.25** | 25.19 |

### 5.1. 4D Occupancy Forecasting

4D occupancy forecasting is a critical perception task in autonomous driving that predicts future occupancy states of the surrounding environment in a spatiotemporal volume (Liu et al., 2025). This task requires modeling complex scene dynamics, making it an ideal testbed for evaluating generative model distillation under real-world constraints.

We conduct experiments on OccFM (Liu et al., 2025) for LiDAR-based 4D occupancy prediction, initializing the teacher, auxiliary, and student models from the same pretrained checkpoint. Experiments are conducted on the nuScenes dataset (Caesar et al., 2020), with performance measured by IoU and mean IoU (mIoU) (Song et al., 2017). We compare against advanced baselines, including Consistency Models (Song et al., 2023), Align Your Flow (AYF) (Sabour et al., 2025), Diff-Instruct (Luo et al., 2023) (VSD-based), and SOTA distillation methods such as DMD2 (Yin et al., 2024a) and SenseFlow (Ge et al., 2026).

Table 1 presents a comprehensive comparison of MFD against the teacher model and baseline distillation methods. Compared with the teacher model with 10-steps sampling, the proposed MFD achieves remarkable distillation quality, losing only **1.2%** of the teacher's IoU (37.07% vs. 37.52%) and **4.4%** of its mIoU (27.25% vs. 28.49%) while increasing the sampling speed **tenfold** ( reducing NFE from 10 to 1) and increasing the FPS from 12.72 to 25.21. Moreover, compared to teacher models with fewer sampling steps, MFD achieves higher generation quality.

Compared with other distillation methods, MFD achieves the optimal performance across all metrics. These performance gaps highlight fundamental limitations of existing methods. VSD suffers from high-variance gradients due to its reliance on single-timestep velocity estimates, which are noisy in high-dimensional latent spaces and fail to capture the global trajectory structure of flow matching. DMD2 and SenseFlow introduce adversarial discriminators, leading to training instability and strong hyperparameter sensitivity, especially when adapting to flow matching models with

Using Equation (20), the student model can be optimized such that its single-step generation distribution closely matches the teacher's multi-step generation distribution. For more discussion on the rationality and theoretical scope of omitting the Jacobian term, please refer to Section B.6. The complete training algorithm of Mean Flow Distillation is summarized in Algorithm 1. Following the TTUR strategy (Heusel et al., 2017) and existing distribution distillation methods (Luo et al., 2023; Yin et al., 2024a), we set the alternating update ratio between the auxiliary model and the student model to 10:1, ensuring that the auxiliary model can accurately track the evolving student distribution before each student update. Measured wall-clock and memory costs are reported in Section 5.3.

## 5. Experiments

In this section, we conduct a series of experiments across two challenging scenarios, 4D occupancy forecasting for autonomous driving (Section 5.1) and text-to-image generation (Section 5.2), to evaluate the effectiveness of MFD. Detailed hyperparameters and training configurations are provided in Section A.3.

*Table 2.* Ablation on ODE solver step size for MFD.

| ODE Solver Step Size | IoU $\uparrow$ (%) | mIoU $\uparrow$ (%) |
|---|---|---|
| 0.001 | 36.28 | 25.91 |
| 0.005 | 37.12 | 27.14 |
| 0.02 | **37.07** | **27.25** |
| 0.1 | 36.82 | 26.47 |
| 0.5 | 36.74 | 26.54 |

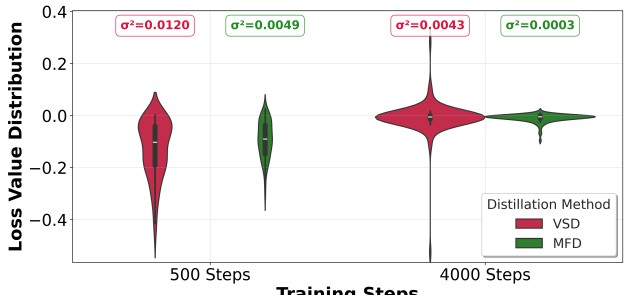

*Figure 2.* Loss distribution between MFD and Diff-Instruct on text-to-image generation at training step 500 and 4000. MFD consistently exhibits a narrower distribution with lower variance, empirically confirming the variance reduction property proven in Section C.3.

different loss landscapes. Moreover, both still depend on instantaneous score matching, which struggles to model long-horizon flow dynamics. Consistency-based methods, including CD and AYF, impose sample-level self-consistency constraints that prove overly restrictive for complex multi-modal data; MFD outperforms AYF by 4.69 IoU and 5.56 mIoU. In contrast, MFD matches time-integrated mean flows at the distribution level, yielding low-variance gradients that preserve generation quality while ensuring stable training. All measured single-step methods achieve similar inference speeds ($\approx$25 FPS), indicating that although MFD introduces an auxiliary flow model $v_\phi$ during training, it incurs negligible inference overhead. Qualitative visualizations are provided in Figure 5 in the appendix.

To investigate the sensitivity of MFD to the numerical accuracy of mean flow computation, we vary the ODE solver step size in $\Phi(v, X_s, t, s)$. Smaller steps reduce discretization error but increase training cost. Results in Table 2 reveal an interesting non-monotonic trend. Very small steps (0.001) hurt performance due to accumulated numerical errors. Moderate steps (0.005–0.02) perform best, with 0.02 achieving the highest mIoU, indicating that a coarser integration provides more robust supervision by averaging noisy velocities. Large steps (0.1–0.5) degrade performance, as too few integration steps fail to capture trajectory curvature. As $\Delta\tau$ approaches $(t - s)$, MFD collapses to a single-step, VSD-like form, losing the variance-reduction benefit of temporal integration. In practice we adopt $0.02$.

**5.2. Text-to-Image Generation**

Text-to-image generation is a fundamental task in generative modeling that synthesizes images from natural language descriptions. Its high-dimensional output space makes it an ideal benchmark for evaluating distillation methods.

We conduct experiments on SANA 1.6B (Xie et al., 2025), a state-of-the-art flow matching model for text-to-image generation, where the teacher model, auxiliary flow model, and student generator are all initialized from the pretrained SANA checkpoint. We train and evaluate on the LAION-aesthetic-6.5+ (Schuhmann et al., 2022) dataset using five complementary metrics: Aesthetic Score (Schuhmann et al., 2022) for visual appeal, PickScore (Kirstain et al., 2023) for human preference prediction, HPSv2 (Wu et al., 2023)

for aesthetic alignment, ImageReward (Xu et al., 2023) for overall quality and text-image alignment, and CLIP Score (Hessel et al., 2021) for semantic alignment. We additionally report FID with DINOv2 features (Oquab et al., 2024) against teacher-generated reference samples to measure distributional proximity. We compare MFD against five representative distillation methods: Diff-Instruct (Luo et al., 2023), Consistency Distillation (Song et al., 2023), AYF (Sabour et al., 2025), DMD2 (Yin et al., 2024a), and SenseFlow (Ge et al., 2026).

Table 3 presents a comprehensive comparison on text-to-image generation. MFD achieves the best Aesthetic Score, ImageReward, CLIP Score, and FID(DINOv2), while matching the strongest baselines on PickScore and HPSv2 within a small margin. Compared with AYF, MFD improves all reported metrics and reduces FID from 43.68 to 33.28, indicating that distribution-level mean-flow alignment provides stronger few-step distillation than sample-level consistency constraints. DMD2 and SenseFlow achieve slightly higher PickScore or HPSv2, but they incorporate additional components such as perceptual losses or adversarial training (GAN) (Goodfellow et al., 2020). In contrast, MFD focuses purely on distribution matching through mean flow alignment as a foundational distillation approach, and can be combined with perceptual losses (Johnson et al., 2016) or adversarial objectives in future work. Coverage-oriented CIFAR-10 precision/recall results are provided in Section D.1.

To empirically validate our theoretical claim that MFD exhibits lower gradient variance than VSD (Section C.3), we conduct a variance analysis experiment on the text-to-image generation task. We perform snapshot analysis at training steps 500 and 4000, measuring student loss distributions by freezing student model parameters and sampling diverse batches. The loss is defined as $\mathbb{E}_{s,t,X_0\sim\mathbb{B}}\left[X_s^\top \cdot \Delta(X_s)\right]$ following Eq (19) and it converges to zero as the training proceeds. Figure 2 presents violin plots comparing the loss distributions of MFD and Diff-Instruct at these two rep-

*Table 3.* Comparison with baselines on text-to-image generation. FID is computed with DINOv2 features (Oquab et al., 2024) against 5,000 teacher-generated reference samples.

| Model | NFE ↓ | Aesthetic Score ↑ | PickScore ↑ | HPSv2 ↑ | ImageReward ↑ | CLIP Score ↑ | FID(DINOv2) ↓ |
|---|---|---|---|---|---|---|---|
| SANA (Teacher) | 20 | 6.335 | 0.2021 | 0.2417 | 0.8363 | 30.717 | – |
| SANA (Teacher) | 10 | 6.297 | 0.2009 | 0.2388 | 0.7788 | 30.508 | – |
| SANA (Teacher) | 4 | 5.976 | 0.1872 | 0.2173 | -0.1008 | 28.251 | – |
| AYF (Sabour et al., 2025) | 4 | 6.231 | 0.1963 | 0.2298 | 0.7022 | 29.176 | 43.68 |
| Diff-Instruct | 4 | 6.453 | 0.1985 | 0.2434 | 0.7677 | 29.393 | 37.55 |
| CD | 4 | 6.242 | 0.1918 | 0.2267 | 0.6734 | 28.925 | 45.97 |
| DMD2 | 4 | 6.495 | **0.2004** | 0.2439 | 0.8017 | 29.240 | 36.13 |
| SenseFlow | 4 | 6.482 | 0.1998 | **0.2445** | 0.7925 | 29.321 | 35.97 |
| **MFD (Ours)** | 4 | **6.541** | 0.2002 | 0.2442 | **0.8483** | **29.430** | **33.28** |

resentative training stages. At both early (step 500) and late (step 4000) training stages, MFD exhibits consistently narrower and more concentrated loss distributions, empirically confirming the variance reduction property. Additionally, we compare the training loss trajectories shown in Figure 6 (Section D.5), where MFD demonstrates significantly smoother convergence with substantially reduced loss fluctuations throughout the entire training process, further validating that the temporal integration in mean flow matching effectively suppresses optimization noise.

Similarly, we investigate the sensitivity of MFD to the numerical accuracy of mean flow computation in the text-to-image generation setting. Table 7 reports the performance of MFD under different ODE solver step sizes, which control the discretization granularity used in mean flow estimation. The ablation study on ODE solver step size reveals consistent trends with the 4D occupancy forecasting experiments. Excessively small step sizes result in degraded performance, likely due to the accumulation of numerical errors over a large number of integration steps. Due to the length limitation, the detailed analysis is shown in Section D.4. Empirically, we recommend the ODE solver step size be set to 0.02 or 0.03. Additionally, we present qualitative comparisons of generated images in Figure 8, demonstrating the visual quality of MFD-distilled models with 4-step inference.

### 5.3. Training Cost and Practical Overhead

The auxiliary flow model $v_\phi$ is discarded at inference, so MFD incurs no extra inference cost (cf. Table 1). Implemented with LoRA (rank 64) on SANA 1.6B, it adds only ∼34M trainable parameters (<2.2% of the backbone). Table 4 compares wall-clock cost and peak GPU memory: MFD (∼25 GPU hours, ∼20 GB on one RTX PRO 6000) is comparable to VSD while using substantially less memory than DMD2, which additionally trains a discriminator. The 10:1 update ratio does not entail a 10-fold cost increase, because the dominant computation is the mean-flow ODE integration during student updates, while the auxiliary model is updated via lightweight adapters. CD is cheaper but at the expense of clearly inferior quality. The training overhead is

*Table 4.* Training cost comparison for text-to-image distillation on SANA 1.6B. All measurements use one NVIDIA RTX PRO 6000 GPU.

| Method | GPU hours | Peak memory (GB) |
|---|---|---|
| Diff-Instruct (VSD) | ∼22 | ∼20 |
| CD | ∼4 | ∼15 |
| DMD2 | ∼30 | ∼50 |
| MFD (ours) | ∼25 | ∼20 |

a one-time distillation cost with zero impact on deployment.

## 6. Conclusion

In this paper, we presented Mean Flow Distillation (MFD), a robust framework that addresses the high gradient variance inherent in score-based distillation by aligning time-integrated average velocity fields. Theoretically grounded in our Mean Flow Matching Theorem, MFD functions as a temporal low-pass filter, suppressing optimization noise to ensure global trajectory consistency. Extensive experiments on 4D occupancy forecasting and text-to-image generation demonstrate that MFD consistently outperforms state-of-the-art baselines, achieving high-fidelity single-step generation with superior training stability and distributional fidelity. While our current approach relies on an auxiliary flow model during training, it adds no inference overhead; future work will further reduce its training cost and broaden the framework to additional flow matching domains.

## Acknowledgements

This paper is funded by Provincial Key Research and Development Plan of Zhejiang Province under No. 2024C01250(SD2), National Natural Science Foundation of China under No. 62576306 and the Fundamental Research Funds for the Zhejiang Provincial Universities under No. 226-2025-00206. We thank anonymous reviewers for their insightful comments and constructive suggestions, which significantly improved the quality of this work.

## Impact Statement

This paper presents work whose goal is to advance the field of Machine Learning, specifically in accelerating generative models through flow matching distillation. The primary societal impact of our work lies in democratizing access to high-quality generative AI by significantly reducing computational costs. By enabling single-step generation with minimal quality loss, MFD can make powerful text-to-image and 4D perception models more accessible to researchers and practitioners with limited computational resources, potentially fostering broader innovation in creative industries and autonomous systems.

However, we acknowledge potential dual-use concerns. Accelerated generative models could facilitate the rapid creation of synthetic media, which may be misused for generating misleading or harmful content. We encourage practitioners deploying MFD-distilled models to implement appropriate safeguards, including content filtering, watermarking, and provenance tracking mechanisms. In the context of autonomous driving applications, while our method improves efficiency, thorough safety validation and testing remain essential before real-world deployment. Overall, we believe the benefits of improved accessibility and efficiency outweigh the risks when accompanied by responsible deployment practices.

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

# A. Experiment protocol

## A.1. Dataset setup

nuScenes (Caesar et al., 2020) is a large-scale multimodal perception dataset for autonomous driving, released by Motional (formerly nuTonomy), and represents the first public dataset to systematically provide a full-suite sensor configuration for autonomous vehicles. Collected in real-world urban driving scenarios, nuScenes comprises synchronized data from six surround-view cameras, one 32-beam LiDAR, five millimeter-wave radars, along with high-precision GPS and IMU measurements, enabling multi-view and multimodal modeling in complex traffic environments. Compared to earlier datasets such as KITTI, nuScenes offers substantially broader sensor coverage as well as significant improvements in data scale and annotation density. It contains over seven times more object-level annotations than KITTI and provides long-term temporal sequences with fine-grained 3D object labels, attributes, and motion states. Owing to its rich multimodal temporal data and high-quality 3D annotations, nuScenes has become a widely adopted benchmark for tasks such as 4D occupancy forecasting, temporal scene understanding, and world modeling.

LAION-aesthetic-6.5+ (Schuhmann et al., 2022) is a high-quality subset of the LAION-5B dataset, specifically curated for training and evaluating text-to-image generation models. LAION (Large-scale Artificial Intelligence Open Network) is one of the largest publicly available image-text pair datasets, collected from the public web through Common Crawl. The aesthetic-6.5+ subset applies an aesthetic quality filter based on the LAION aesthetic predictor, retaining only images with predicted aesthetic scores of 6.5 or higher (on a scale from 1 to 10). This filtering process ensures that the dataset contains visually appealing, high-quality images with good composition, color harmony, and overall aesthetic appeal, making it particularly suitable for training generative models that aim to produce visually pleasing outputs. Each sample in LAION-aesthetic-6.5+ consists of an image paired with its corresponding alt-text caption scraped from the web, providing natural language descriptions that range from short phrases to detailed sentences. The dataset covers an extremely diverse range of visual concepts, including everyday objects, natural scenes, portraits, abstract art, architectural structures, and various artistic styles. This diversity in both visual content and textual descriptions enables text-to-image models to learn robust cross-modal alignment and generate high-fidelity images across a wide spectrum of prompts. Due to its large scale, high aesthetic quality, and rich semantic diversity, LAION-aesthetic-6.5+ has become a de facto standard benchmark for training and fine-tuning state-of-the-art text-to-image generation models, including Stable Diffusion, DALL-E variants, and flow-based generative models.

## A.2. Evaluation metrics

Intersection over Union (IoU) (Song et al., 2017) is a classical evaluation metric that measures the degree of overlap between model predictions and ground-truth annotations, and is widely used in tasks such as object detection, semantic segmentation, and occupancy prediction. It is defined as the ratio between the intersection and the union of the predicted region and the ground-truth region, taking values in the range $[0, 1]$, where higher values indicate better agreement with the ground truth. In 3D or 4D scene understanding tasks, IoU can be naturally extended to voxel-level or spatiotemporal volume-level IoU, enabling the evaluation of a model's ability to capture spatial structures and their temporal evolution. As such, IoU serves as a fundamental metric for assessing prediction quality at the level of individual classes or samples.

Mean Intersection over Union (mIoU) aggregates IoU scores across multiple semantic classes or time steps to provide a holistic measure of overall model performance. Specifically, mIoU is typically computed by first evaluating IoU for each semantic category (or occupancy state) independently and then taking the arithmetic mean over all categories. This averaging strategy prevents the evaluation from being dominated by classes with a large number of samples. Compared to a single IoU score, mIoU more effectively reflects a model's class-wise balance and robustness, and is therefore widely adopted as a core evaluation metric in semantic segmentation, occupancy grid prediction, and 4D occupancy forecasting tasks.

Aesthetic Score (Schuhmann et al., 2022) is an automated image quality assessment metric developed as part of the LAION project to predict the aesthetic appeal of generated images. It is trained to predict human aesthetic preferences on a scale from 1 to 10, with higher scores indicating more visually pleasing images in terms of composition, color harmony, lighting, and overall artistic quality. The predictor is trained on a large dataset of images with human-annotated aesthetic ratings, enabling it to serve as a proxy for human aesthetic judgment. This metric is particularly useful for evaluating text-to-image generation models, as it captures subjective visual quality aspects that go beyond simple semantic alignment or technical correctness.

PickScore (Kirstain et al., 2023) is a human preference prediction metric specifically designed for text-to-image generation evaluation. Unlike traditional metrics that focus on individual image attributes, PickScore is trained to predict which image humans would prefer when presented with pairs of generated images for the same text prompt. The metric is based on a vision-language model fine-tuned on a large-scale dataset of human preference comparisons collected through crowdsourcing. PickScore provides a scalar value that correlates with human preference rankings, making it a reliable indicator of generation quality from a human-centered perspective. This metric is particularly valuable for assessing the overall appeal and naturalness of generated images beyond pixel-level or feature-level similarity measures.

HPSv2 (Human Preference Score v2) (Wu et al., 2023) is an advanced human preference prediction metric that represents the second generation of human preference scoring systems for text-to-image generation. Building upon the original HPS framework, HPSv2 incorporates improved training strategies and a more diverse preference dataset to better capture the nuanced aspects of human aesthetic judgment. The metric evaluates multiple dimensions of image quality including aesthetic appeal, prompt alignment, photorealism, and artistic coherence. HPSv2 demonstrates strong correlation with human evaluations across diverse generation scenarios and has become widely adopted as a standard benchmark for comparing text-to-image generation models.

ImageReward (Xu et al., 2023) is a comprehensive reward function designed to holistically assess the quality of text-to-image generation by jointly considering multiple aspects including visual fidelity, semantic alignment with text prompts, aesthetic quality, and overall human preference. Unlike single-dimensional metrics, ImageReward is trained using reinforcement learning from human feedback (RLHF) on a large-scale dataset of human annotations that capture diverse quality aspects. The metric provides a unified score that balances technical correctness (e.g., object presence, spatial relationships) with subjective quality judgments (e.g., artistic appeal, naturalness). ImageReward has been shown to correlate strongly with comprehensive human evaluations and is particularly effective for optimizing text-to-image models through reward-based fine-tuning strategies.

CLIP Score (Hessel et al., 2021) measures the semantic alignment between generated images and their corresponding text prompts using the CLIP (Contrastive Language-Image Pre-training) model. CLIP is trained on hundreds of millions of image-text pairs from the internet to learn joint embeddings where semantically related images and texts are mapped to nearby points in a shared representation space. The CLIP Score is computed as the cosine similarity between the CLIP embeddings of a generated image and its text prompt, providing a quantitative measure of how well the image content matches the textual description. Higher CLIP Scores indicate stronger semantic correspondence, capturing whether the generated image contains the correct objects, attributes, actions, and relationships specified in the prompt. While CLIP Score primarily evaluates semantic fidelity rather than aesthetic quality or photorealism, it serves as a fundamental metric for assessing the text-image alignment capability of generation models and has become a standard evaluation tool in the text-to-image generation literature.

Fréchet Inception Distance (FID) measures the distance between feature distributions of generated and reference images by fitting Gaussian distributions to their extracted features. In our text-to-image experiments, we use DINOv2 features (Oquab et al., 2024) and compute FID against teacher-generated reference samples, which directly evaluates how closely the student preserves the teacher distribution. For CIFAR-10 coverage analysis, we additionally report precision and recall (Kynkäänniemi et al., 2019): precision measures sample fidelity by estimating how much of the generated distribution lies on the data manifold, while recall measures coverage by estimating how much of the reference distribution is recovered by the generator.

### A.3. Implementation details

#### A.3.1. 4D Occupancy Forecasting

During the training, the variational autoencoder (VAE) used for latent space encoding is kept frozen with pretrained weights throughout training. Except for the auxiliary model and student model, all other network parameters remain frozen during distillation. The student model is configured as a single-step generator $G_\theta : \mathbb{R}^d \to \mathbb{R}^d$, directly mapping noise to the final occupancy prediction without iterative refinement.

We employ the Adam optimizer with a learning rate of $1 \times 10^{-5}$ for both the auxiliary flow model $v_\phi$ and the student model $G_\theta$, incorporating a linear warmup schedule over the first 100 iterations to stabilize early-stage training. Gradient clipping with a maximum norm of 1.0 is applied. Training is conducted on a single node equipped with 8 NVIDIA RTX 3090 GPUs using distributed data parallel (DDP). We set the batch size to 1 per GPU (effective batch size of 8), with mixed-precision

training (FP16) enabled to accelerate computation and reduce memory consumption using a gradient rescaling factor of 10.0.

Following recent findings in diffusion model training, we adopt a logit-normal biased sampling strategy (Esser et al., 2024) for timesteps $t$ and $s$, which allocates more sampling density to intermediate timesteps where the velocity field exhibits higher variance. When computing the mean flow $\hat{U}(X_s, s, t)$, we employ an adaptive ODE solver with a maximum step size of 0.02 to ensure accurate numerical integration along the velocity field—our ablation study validates this choice as an optimal trade-off between integration accuracy and computational efficiency. During distillation, the teacher model $u^Q$ is evaluated with classifier-free guidance (CFG) enabled at a guidance scale of $w = 2.0$ to enhance conditional generation quality and provide stronger supervision signals, while the auxiliary flow model $v_\phi$ is trained without CFG to directly model the student distribution $P$. We train for a total of 30 epochs on the nuScenes training set (approximately 28,000 annotated keyframes) and use the final checkpoint for evaluation, as we observe minimal overfitting due to the regularization provided by the pretrained initialization and frozen VAE encoder.

### A.3.2. TEXT-TO-IMAGE GENERATION

We train our model on the LAION-aesthetic-6.5+ dataset, using the text prompt portion as training data. To enable efficient fine-tuning, we employ Low-Rank Adaptation (LoRA) with rank $r = 64$ and LoRA alpha $\alpha = 64$. In the SANA 1.6B experiments, the auxiliary model is therefore not a full-parameter copy of the teacher: only approximately 34M LoRA parameters are trainable, which is less than 2.2% of the 1.6B-parameter backbone. We use the 8-bit Adam optimizer (AdamW8bit) with its default parameters for memory-efficient training. The learning rate is set to $1 \times 10^{-4}$ for both the auxiliary flow model $v_\phi$ and the student generator $G_\theta$, without any learning rate scheduling or warmup.

Training is conducted on 1 NVIDIA PRO 6000 GPU, with a batch size of 1 per GPU and no gradient accumulation. We enable mixed-precision training using BF16 to accelerate computation and reduce memory consumption while maintaining numerical stability. Gradient clipping with a maximum norm of 0.5 is applied to prevent gradient explosion. The output image resolution is set to $1024 \times 1024$ pixels to generate high-fidelity images.

For timestep sampling, we adopt a logit-normal biased sampling strategy similar to the 4D occupancy forecasting experiments, which allocates more sampling density to intermediate timesteps where the velocity field exhibits higher variance. When computing the mean flow, we employ an ODE solver with step size $\Delta\tau = 0.03$. The student model is configured for 4-step inference during evaluation. During distillation, the teacher model $u^Q$ is evaluated with classifier-free guidance (CFG) at a guidance scale of $w = 4.5$ to provide strong conditional generation signals, while the student model operates without CFG to directly model the target distribution. We train for a total of 3000 optimization steps and use the final checkpoint for evaluation.

## B. Discussion

### B.1. Relationship to Align Your Flow and Mean Flow

MFD is closely related to recent flow-model acceleration methods, but it occupies a different optimization regime. Align Your Flow (AYF) (Sabour et al., 2025) is a consistency distillation method: it enforces that points along a teacher ODE trajectory map to the same terminal sample. MFD instead performs distribution matching in velocity space by aligning expected average velocity fields. Thus, AYF imposes sample-level flow-map consistency, while MFD imposes distribution-level mean-flow consistency. This distinction matters early in distillation, when the student distribution can be far from the teacher: exact sample-level constraints can be overly restrictive, whereas expected mean-flow matching tolerates imperfect intermediate samples while still guaranteeing $P = Q$ at convergence under Theorem 4.3.

Mean Flow (MF) (Geng et al., 2025a) shares the idea of supervising time-integrated velocities, but the setting and objective are different. MF trains a generative flow model from scratch using data-derived mean velocity targets. MFD distills an already pretrained flow matching teacher into a single-step or few-step generator, and introduces an auxiliary flow model to estimate the velocity field induced by the evolving student distribution. Consequently, MFD is intended as a general acceleration mechanism for existing high-quality FM models rather than a replacement for full generative training.

### B.2. The difference between Mean Flow Distillation and Rectified Reflow

While 2-Rectified Flow (Reflow) (Liu et al., 2023) and Mean Flow Distillation (MFD) share the overarching goal of accelerating sampling, they represent fundamentally different approaches to the problem of flow simplification. In this

*Table 5.* Conceptual comparison between MFD and closely related flow acceleration methods.

| Method | Setting | Alignment target | Main constraint |
|---|---|---|---|
| AYF (Sabour et al., 2025) | Distillation | Flow map | Sample-level consistency |
| MF (Geng et al., 2025a) | Training from scratch | Mean velocity target | Data-derived flow learning |
| MFD (ours) | Distillation | Expected average velocity | Distribution-level matching |

section, we argue that Reflow and MFD operate in distinct regimes. Reflow is a trajectory regularization technique, whereas MFD is a distribution alignment framework. We provide a detailed analysis of why the instantaneous supervision in Reflow remains insufficient for high-fidelity one-step generation on complex manifolds, and how MFD addresses this through temporal integration.

### B.2.1. OPTIMIZATION DYNAMICS

The most important difference lies in what is smoothed during optimization.

**Reflow Smoothes the Target (Label).** The objective of Reflow is to regress the student's instantaneous velocity $v_\theta(X_t, t)$ toward the straight path connecting data pairs $(X_0, X_1)$:

$$\mathcal{L}_{\text{Reflow}} = \mathbb{E}_{t, X_0} \left[ \|v_\theta(X_t, t) - (X_1 - X_0)\|^2 \right]. \tag{21}$$

Here, the target $Y = (X_1 - X_0)$ is indeed a low-variance, "straightened" signal because it represents the integral of the teacher's trajectory, $\int_0^1 u_{\text{teacher}} d\tau$. This effectively averages out the curvature of the teacher flow. However, the gradient of this loss with respect to parameters $\theta$ depends directly on the student's instantaneous output:

$$\nabla_\theta \mathcal{L}_{\text{Reflow}} \propto (v_\theta(X_t, t) - Y) \cdot \nabla_\theta v_\theta(X_t, t). \tag{22}$$

Crucially, if the student network $v_\theta$ exhibits high variance or stochasticity at specific timesteps $t$ due to mini-batch noise or optimization instability, this noise propagates directly into the gradient update. Smoothing the target $Y$ does not mitigate the noise inherent in the estimator $v_\theta(X_t, t)$.

**MFD Smoothes the Estimator (Gradient).** In contrast, MFD defines the loss over the integrated output of the auxiliary student flow $v^P$:

$$\mathcal{L}_{\text{MFD}} \propto \left\| \int_s^t v^P(X_\tau, \tau) d\tau - \int_s^t u^Q(X_\tau, \tau) d\tau \right\|^2. \tag{23}$$

From a signal processing perspective, the integration operator $\mathcal{I}(\cdot) = \int_s^t (\cdot) d\tau$ acts as a temporal low-pass filter on the student's vector field. If we model the student's instantaneous estimation as the true signal plus zero-mean optimization noise, $\hat{v} = v + \xi$, the Reflow gradient variance is bounded by $\text{Var}(\xi)$. The MFD gradient variance, however, is determined by the variance of the integral of $\xi$.

As derived in our theoretical analysis (Appendix C.3), for a discrete approximation with $K$ steps, the variance scales as:

$$\text{Var}(\nabla \mathcal{L}_{\text{MFD}}) \approx \frac{1}{K} \text{Var}(\nabla \mathcal{L}_{\text{Reflow}}). \tag{24}$$

This establishes MFD not merely as an alternative, but as a strictly more stable optimization objective. On complex tasks like 4D occupancy forecasting, where the manifold topology induces high-frequency fluctuations in velocity fields, this factor of $1/K$ variance reduction is the difference between convergence and mode collapse.

### B.2.2. GEOMETRIC INTERPRETATION

Reflow and MFD enforce different geometric constraints on the generative mapping.

Reflow assumes that the optimal transport map can be well-approximated by a velocity field that is constant in time ($v = \text{const}$). While theoretically possible (Monge-Ampère transportation), in practice, learning a perfectly straight flow on high-dimensional, multimodal data is challenging. Residual curvature often remains. Consequently, sampling Reflow with a single step ($NFE = 1$) is an approximation: $X_1 \approx X_0 + v(X_0)$. Any deviation from straightness results in discretization error (truncation error), leading to blurry or artifact-heavy samples (as observed in Table 1 of the main text).

MFD does not inherently force the auxiliary flow to be straight. Instead, it encourages the student generator $G_\theta$ (a one-step mapper) to land on the correct distribution $Q$, as a relaxed target. The auxiliary model $v^P$ and the integral formulation serve as a bridge. By matching the Mean Flow, we enforce:

$$\underbrace{X_0 + \int_0^1 v^P d\tau}_{\text{Student Endpoint}} = \underbrace{X_0 + \int_0^1 u^Q d\tau}_{\text{Teacher Endpoint}}. \tag{25}$$

This ensures that the destination matches the teacher, regardless of the intermediate path complexity. MFD creates a student $G_\theta$ that is explicitly trained to minimize the error of the final sample $X_1$, whereas Reflow minimizes the error of the vector field $v_t$. For one-step generation, minimizing endpoint error (via MFD) is the more direct and theoretically sound objective.

### B.2.3. JUSTIFICATION FOR THE AUXILIARY MODEL

A potential critique of MFD is the extra computational overhead of training an auxiliary flow model $v^P$. We argue this cost is justified and necessary for three reasons:

1. Decoupling Generation and Estimation. The student generator $G_\theta$ is a direct $X_0 \to X_1$ mapper. It does not possess a time-dependent vector field structure required for ODE-based comparison. The auxiliary model $v^P$ acts as a variational proxy, reconstructing the vector field implied by $G_\theta$. This allows us to leverage the robust mathematical framework of Flow Matching to train a model that does not output vector fields (the Student).

2. Zero Inference Overhead. It is crucial to note that the auxiliary model $v^P$ is strictly a training-time artifact. During inference, only $G_\theta$ is used. Thus, MFD achieves the inference efficiency of a GAN (1 step) with the training stability of Flow Matching.

3. Handling Complex Topologies. In our experiments on LiDAR-based 4D occupancy, we observed that direct distillation methods failed to capture the discrete-continuous nature of the data. The auxiliary model provides the necessary capacity to approximate the mean velocity of these complex transitions, preventing the "averaging" artifacts common in simpler regression objectives. We give a mathematical proof of this in Section C.4.

### B.3. A Signal Processing Perspective on Flow Distillation

In this section, we re-examine Mean Flow Distillation through the perspective of signal processing, formally characterizing the proposed method as a temporal low-pass filter applied to the supervision signal. We posit that this spectral filtering property is the primary driver of the observed training stability and variance reduction.

In the context of distillation, the student model $G_\theta$ attempts to align its induced flow with that of a pre-trained teacher $u^Q$. However, the teacher's instantaneous velocity field $u^Q(X_t, t)$ often serves as a noisy proxy for the ideal transport dynamics due to optimization stochasticity and high-frequency curvature in the learned manifold.

We define the supervision signal $\mathbf{S}(t)$ at a trajectory point $X_t$ as a superposition of the true underlying transport semantics $\mathbf{v}^*(t)$ (the signal) and a stochastic error term $\boldsymbol{\xi}(t)$ (the noise):

$$\mathbf{S}(t) = u^Q(X_t, t) = \mathbf{v}^*(t) + \boldsymbol{\xi}(t), \tag{26}$$

where $\boldsymbol{\xi}(t)$ represents high-frequency oscillatory components arising from approximation errors or local geometric perturbations. In standard Instantaneous Velocity Field Matching (analogous to VSD), the optimization objective $\mathcal{L}_{\text{IVF}}$ forces the student to minimize the discrepancy at every instant $t$:

$$\min_\theta \mathbb{E}_t \left[ \| v^P(X_t, t) - (\mathbf{v}^*(t) + \boldsymbol{\xi}(t)) \|_2^2 \right]. \tag{27}$$

From a signal processing standpoint, this objective functions as an all-pass filter. The student is penalized for failing to reproduce the noise term $\boldsymbol{\xi}(t)$, leading to high-variance gradients and overfitting to local irregularities that are irrelevant to the global transport map.

MFD fundamentally alters this dynamic by shifting the alignment target from the instantaneous velocity to the Average Velocity Field (AVF). Recall our definition of the AVF over the interval $[s, t]$:

$$U(X_s, s, t) = \frac{1}{t - s} \int_s^t u^Q(\psi_\tau(X_0), \tau) \, d\tau. \tag{28}$$

Applying the signal decomposition to the AVF yields:

$$U(X_s, s, t) = \underbrace{\frac{1}{\Delta t} \int_s^t \mathbf{v}^*(\tau) \, d\tau}_{\text{Smoothed Signal}} + \underbrace{\frac{1}{\Delta t} \int_s^t \boldsymbol{\xi}(\tau) \, d\tau}_{\text{Partially Filtered Noise}}, \tag{29}$$

where $\Delta t = t - s$. The operation $\frac{1}{\Delta t} \int_s^t (\cdot) d\tau$ corresponds to a temporal boxcar filter (or moving average). In the frequency domain, the magnitude response of such a filter is given by the sinc function $|\text{sinc}(f\Delta t)|$, which decays as $1/f$. Consequently, the integration operator acts as a low-pass filter, significantly attenuating the high-frequency noise components in $\boldsymbol{\xi}(t)$ while preserving the low-frequency trends of the transport trajectory $\mathbf{v}^*(t)$.

This spectral interpretation explains why MFD is robust to local perturbations. By supervising the student with the integral of the velocity field—effectively the displacement vector—we enforce global trajectory consistency (phase alignment) rather than local tangent alignment. The student learns to match the global displacement rather than the volatile instantaneous velocity.

### B.4. Future Research Directions

The success of Mean Flow Distillation (MFD) suggests a fundamental shift in how we approach generative model acceleration: moving from instantaneous velocity matching to time-integrated mean flow alignment significantly reduces gradient variance and improves training stability. This integral-based supervision effectively functions as a temporal low-pass filter against optimization noise, opening the door to multi-scale temporal distillation strategies. From a geometric perspective, MFD's emphasis on temporal trajectory implies that preserving the properties of the transport path is more critical than precise pointwise alignment. This insight paves the way for extending flow distillation to non-Euclidean manifolds or the spectral domain, where explicit trajectory regularization can better handle complex data topologies. Ultimately, MFD demonstrates that robust distillation requires looking beyond the teacher's instantaneous output to capture the cumulative dynamics of where the teacher leads over time.

### B.5. Discussion on Error Accumulation in Temporal Integration

One intuitive perspective on why Mean Flow Distillation provides stronger supervision than instantaneous velocity matching is the error accumulation effect during temporal integration. This viewpoint suggests that if the student's velocity field deviates from the teacher's at any point along a trajectory, these local deviations compound when integrated over the time interval $[s, t]$, resulting in larger discrepancies in the average velocity field. In this sense, temporal integration acts as an amplification mechanism that makes even small instantaneous velocity errors more detectable and correctable.

Consider a particle moving along a flow trajectory. If at each infinitesimal time step $d\tau$, the student's velocity $v^P(X_\tau, \tau)$ differs from the teacher's velocity $u^Q(X_\tau, \tau)$ by a small amount $\delta v(\tau)$, then over the interval $[s, t]$, the cumulative displacement error becomes:

$$\Delta_{\text{cumulative}} = \int_s^t \delta v(\tau) \, d\tau = \int_s^t \left( v^P(X_\tau, \tau) - u^Q(X_\tau, \tau) \right) d\tau \tag{30}$$

This cumulative error is reflected in the mean velocity discrepancy:

$$U^P(X_s, s, t) - U^Q(X_s, s, t) = \frac{1}{t - s} \int_s^t \left( v^P(X_\tau, \tau) - u^Q(X_\tau, \tau) \right) d\tau \tag{31}$$

From this perspective, the temporal integration converts pointwise errors into integrated displacement errors, which could be viewed as providing a stronger supervision signal that accumulates information about the velocity field mismatch over the entire time interval rather than at a single instant.

**Relationship to Variance Reduction.** While this error accumulation perspective offers an intuitive explanation for why matching mean velocities might be more effective than matching instantaneous velocities, it is important to note that this viewpoint focuses on the magnitude of the supervision signal under the assumption that the student model systematically deviates from the teacher. In practice, the primary challenge in distillation is not necessarily the magnitude of the error signal, but rather the variance and noise in gradient estimation caused by stochastic factors such as mini-batch sampling, limited model capacity, and numerical approximation errors.

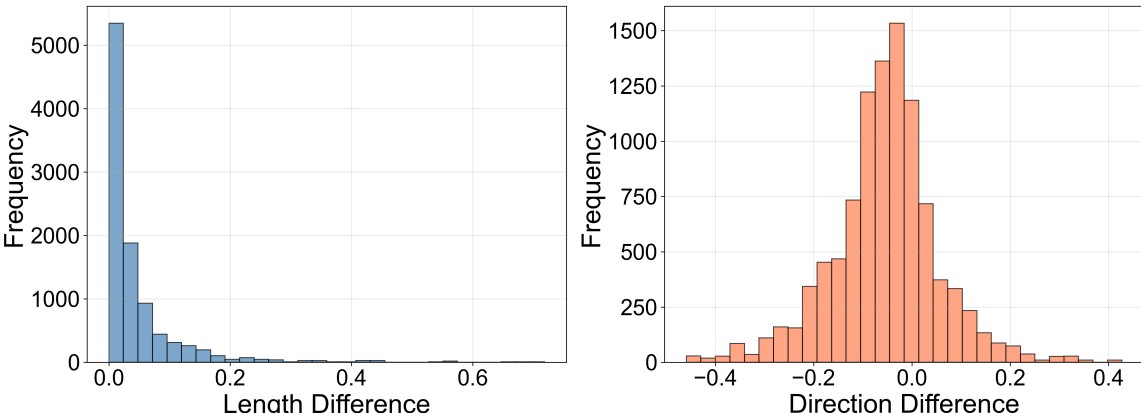

*Figure 3.* Comparison of gradient differences when computing vs. omitting the Jacobian term during backpropagation to $X_s$ over 10,000 training steps. Left: Length difference histogram showing the absolute difference in gradient magnitudes $|\|\nabla_{X_s}^{\text{full}}\| - \|\nabla_{X_s}^{\text{approx}}\||$. Right: Direction difference histogram showing the cosine similarity between normalized gradient vectors. The results reveal that while gradient magnitudes remain similar, the directions diverge significantly when including the full Jacobian computation.

Our theoretical analysis in Section C.3 demonstrates that the key advantage of MFD lies in its variance reduction property: by averaging instantaneous velocity estimates over multiple integration steps, the temporal integration acts as a low-pass filter that suppresses high-frequency optimization noise, yielding gradient estimates with variance scaling as $\frac{1}{K}$ (where $K$ is the number of integration steps) compared to instantaneous matching. This variance reduction is the mathematically rigorous mechanism underlying MFD's improved training stability and convergence.

In summary, the error accumulation and variance reduction perspectives are complementary rather than contradictory:

- **Error accumulation** provides an intuitive geometric interpretation: temporal integration converts local velocity mismatches into global displacement errors, potentially making the supervision signal more informative about trajectory-level alignment.

- **Variance reduction** provides a rigorous statistical analysis: temporal integration averages out stochastic noise across multiple time steps, yielding more stable and reliable gradient estimates that facilitate convergent optimization.

While we find the error accumulation viewpoint conceptually appealing, we have not yet identified a suitable mathematical or experimental framework to rigorously validate and quantify this effect independently of the variance reduction mechanism. Future work could explore decomposing these two effects—perhaps through carefully controlled synthetic experiments where the noise level and systematic bias of velocity field estimates can be independently manipulated—to better understand their relative contributions to MFD's effectiveness.

### B.6. Discussion on the rationality of omitting the Jacobian term

In the derivation of Mean Flow Distillation, when computing the gradient with respect to the student generator parameters $\theta$, the exact mathematical formulation would involve backpropagating through the ODE integration process, which theoretically requires computing the Jacobian matrices of both the auxiliary flow model $v_\phi$ and the teacher model $u^Q$. However, in our practical implementation, we treat these Jacobian terms as identity matrices, effectively stopping the gradient flow through the auxiliary and teacher models when computing $\nabla_\theta \mathcal{L}_{MFD}$. This approximation, while mathematically inexact, is critical for achieving stable and high-quality distillation results.

To empirically investigate the impact of this approximation, we conducted a comparative analysis during the text-to-image generation experiments (SANA 1.6B (Xie et al., 2025)) over the first 10,000 training steps. We computed the gradients with respect to the intermediate variable $X_s$ under two scenarios: (1) with full Jacobian computation (backpropagating through both auxiliary and teacher models), and (2) with the identity matrix approximation (stopping gradients at the auxiliary and teacher models). Figure 3 presents histograms comparing these two gradient computation strategies along two critical dimensions.

The left panel of Figure 3 shows the length difference histogram, which measures the absolute difference between the L2

norms of the two gradient vectors: $|\|\nabla_{X_s}^{\text{full}}\| - \|\nabla_{X_s}^{\text{approx}}\||$. This metric evaluates whether the gradient magnitudes remain consistent across the two computation strategies. The right panel shows the direction difference histogram, which computes the cosine similarity between the normalized gradient vectors, providing a measure of directional alignment in the range $[-1, 1]$, where values close to 1 indicate strong directional agreement.

The empirical results reveal a striking phenomenon: while the gradient magnitudes remain relatively consistent (left panel shows most differences concentrated near zero), the gradient directions diverge dramatically (right panel shows a broad distribution with many samples exhibiting low or even negative cosine similarity). This directional inconsistency explains why training with full Jacobian computation fails to produce reasonable, high-quality results in our experiments. We hypothesize that this directional distortion arises because backpropagating through both the auxiliary flow model and the teacher model introduces substantial noise and complex higher-order dependencies that corrupt the gradient signal. The ODE integration process involves multiple sequential function evaluations, and backpropagating through this entire computational graph amplifies approximation errors and numerical instabilities, ultimately distorting the gradient direction away from the true descent direction toward the optimal student distribution.

In contrast, the identity matrix approximation, stopping gradients at the auxiliary and teacher models, provides a cleaner, more direct supervision signal. This approximation aligns with the motivation behind score distillation methods: the teacher model serves as a fixed evaluator that provides supervision signals, rather than a differentiable component of the student's training objective. By treating the mean flow computed from the teacher and auxiliary models as a fixed target, we avoid the gradient distortion caused by backpropagating through complex, potentially ill-conditioned neural ODEs. This pragmatic approximation strategy has precedent in the distillation literature. Score Distillation Sampling (SDS) (Poole et al., 2023) and its variants similarly stop gradients through the teacher diffusion model, treating the teacher's score predictions as fixed supervision signals rather than learnable components.

We do not claim a tight theoretical bound on the error introduced by omitting the Jacobian term. Deriving such a bound for neural ODE distillation remains an open problem shared by SDS, VSD, DMD, and DMD2-style objectives, which also treat teacher predictions as fixed guidance targets. Importantly, the approximation affects the optimization trajectory rather than the fixed point characterized by Theorem 4.3: if the expected average velocity fields are matched exactly, the Mean Flow Matching Theorem still implies $P = Q$. The approximation is therefore best understood as an empirically necessary optimization device for reaching a stable solution in high-dimensional models, not as a change to the distributional alignment condition itself.

In summary, while the identity matrix approximation of the Jacobian term is mathematically inexact, it is a principled and empirically validated engineering choice that prevents gradient distortion, stabilizes optimization, and enables MFD to achieve state-of-the-art distillation performance. The stark directional divergence observed in Figure 3 provides compelling evidence that omitting the Jacobian computation is not merely a computational convenience, but a necessary design decision for robust flow matching distillation.

## C. Theoretical Demonstration

### C.1. The Proof of Theorem 4.1 (Instantaneous Velocity Field Matching)

**Theorem Statement:** Let $u^Q$ and $v^P$ be the instantaneous velocity fields corresponding to the teacher distribution $Q$ and the student distribution $P$, respectively. Assume both velocity fields satisfy Lipschitz continuity. Let $D(\cdot, \cdot)$ denote a Bregman divergence on $\mathbb{R}^d$ satisfying strict positive definiteness ($D(x, y) = 0 \iff x = y$). If the expected Bregman divergence between the instantaneous velocity fields is zero:

$$\mathcal{L}_{IVF} = \mathbb{E}_{t \sim \mathcal{U}[0,1], X_t \sim p_t^P} \left[ D\left( v^P(X_t, t), u^Q(X_t, t) \right) \right] = 0, \tag{32}$$

then the student distribution coincides with the teacher distribution, i.e., $P = Q$.

**Proof:** Since $\mathcal{L}_{IVF}$ is the expectation of a non-negative term and equals zero, the integrand must be zero almost everywhere. Given that: (1) $D(x, y) \geq 0$; (2) $\mathcal{U}[0, 1]$ has full support; (3) $p_t^P$ is the probability density induced by the flow $\psi_t^P$; and (4) the velocity fields satisfy Lipschitz continuity, we conclude that for all $t \in [0, 1]$ and all $x$ in the support of $p_t^P$:

$$v^P(x, t) = u^Q(x, t) \tag{33}$$

Consider any initial point $X_0 \sim p_0$. The trajectory under the student flow satisfies:

$$\frac{d}{dt}\psi_t^P(X_0) = v^P(\psi_t^P(X_0), t), \quad \psi_0^P(X_0) = X_0 \tag{34}$$

Since $v^P(x, t) = u^Q(x, t)$ everywhere on the support, this trajectory also satisfies the teacher ODE:

$$\frac{d}{dt}\psi_t^P(X_0) = u^Q(\psi_t^P(X_0), t) \tag{35}$$

By the Picard-Lindelöf theorem (uniqueness of ODE solutions under Lipschitz conditions), we have $\psi_t^P(X_0) = \psi_t^Q(X_0)$ for all $t \in [0, 1]$. Since this holds for any $X_0 \sim p_0$, the induced distributions satisfy $P = (\psi_1^P)_{\#}p_0 = (\psi_1^Q)_{\#}p_0 = Q$.

### C.2. The Proof of Theorem 4.3 (Mean Flow Matching Theorem)

**Theorem Statement:** Let $U^Q(z, s, t)$ and $U^P(z, s, t)$ be the average velocity fields corresponding to the distributions $Q$ and $P$, respectively. Taking $U^P$ as an example, it satisfies the following integral relationship for $0 \le s < t \le 1$:

$$(t - s)U^P(z_s, s, t) = \int_s^t v^P\left(\psi_\tau^P\left((\psi_s^P)^{-1}(z_s)\right), \tau\right) d\tau. \tag{36}$$

Intuitively, the AVF describes the mean displacement vector of a particle moving from position $z_s$ to $z_t$ over the time interval $[s, t]$, i.e., $z_t = z_s + (t - s)U^P(z_s, s, t)$.

Let $D(\cdot, \cdot)$ denote a Bregman divergence on $\mathbb{R}^d$ satisfying strict positive definiteness ($D(x, y) = 0 \iff x = y$). If the expected Bregman divergence between the AVF of the auxiliary flow $P$ and the teacher flow $Q$ is zero:

$$\mathcal{L}_{MFD} = \mathbb{E}_{s, t \sim p(s, t), X_s \sim p_s^P} \\ \left[D\left(U^P(X_s, s, t), U^Q(X_s, s, t)\right)\right] = 0, \tag{37}$$

then the student distribution coincides with the teacher distribution, i.e., $P = Q$. Here, $p(s, t)$ denotes a distribution with full support on the time domain, and the velocity fields are assumed to satisfy Lipschitz continuity.

**Proof:** We provide a proof by contradiction to demonstrate that minimizing the Mean Flow Distillation loss implies matching the instantaneous velocity fields.

**Assumption for Contradiction:** Assume that the student instantaneous velocity field $v^P$ does not match the teacher instantaneous velocity field $u^Q$ somewhere. That is, there exists a set $\Omega$ with positive measure under the measure induced by the flow, such that for all $(x, s) \in \Omega$:

$$v^P(x, s) \ne u^Q(x, s) \tag{38}$$

Given that the vector fields satisfy Lipschitz continuity conditions, if they differ at a point, they must differ in a local neighborhood. Thus, there exists a continuous region (a ball of radius $\epsilon > 0$) around some point $(x^*, s^*) \in \Omega$ where the inequality holds strictly. Let us denote this region as $\mathcal{B}_\epsilon$.

We now examine the Average Velocity Field $U(x, s, t)$, which relates to the instantaneous field via integration:

$$U(x, s, t) = \frac{1}{t - s} \int_s^t v(\psi_\tau(x_0), \tau) d\tau \tag{39}$$

If $v^P(x, \tau) \ne u^Q(x, \tau)$ consistently within the measurable region $\mathcal{B}_\epsilon$, for any starting time $s$ inside this region, there exists a sufficiently small $\delta > 0$ such that for all $t \in (s, s + \delta]$, the integral averages (the mean flows) must also differ:

$$U^P(x, s, t) \ne U^Q(x, s, t) \tag{40}$$

Intuitively, two continuous functions that are strictly separated at a point cannot have identical averages over an arbitrarily small interval around that point.

**Deriving the Contradiction:** The strict positive definiteness of the Bregman divergence states that $D(x, y) \ge 0$ and $D(x, y) = 0 \iff x = y$. Consequently, in the region identified above where $U^P \ne U^Q$, we have:

$$D\left(U^P(x, s, t), U^Q(x, s, t)\right) > 0 \tag{41}$$

Since the sampling distribution $p(s, t)$ has full support over the domain $0 \leq s < t \leq 1$, it assigns non-zero probability mass to the interval $(s, s + \delta]$ within the region $\mathcal{B}_\epsilon$. Therefore, when integrating the loss function, we are integrating a strictly positive term over a set of positive measure:

$$\mathcal{L}_{MFD} \geq \int_{\mathcal{B}_\epsilon \cap \{t \in (s, s+\delta]\}} p(s, t) p_s^P(x) \underbrace{D\left(U^P, U^Q\right)}_{>0} dx \, ds \, dt > 0 \tag{42}$$

This implies $\mathcal{L}_{MFD} > 0$, which directly contradicts the premise that $\mathcal{L}_{MFD} = 0$.

**Conclusion:** The assumption that $v^P \neq u^Q$ on a set of positive measure must be false. Therefore, we conclude that:

$$v^P(x, s) = u^Q(x, s) \quad \text{almost everywhere} \tag{43}$$

By invoking Theorem 4.1 (Instantaneous Velocity Field Matching), the equality of instantaneous velocity fields implies the alignment of the generated distributions, i.e., $P = Q$.

### C.3. MFD Has Lower Variance

In this part, through theoretical analysis, we demonstrate that Mean Flow Distillation (MFD) exhibits significantly lower gradient estimation variance than Variational Score Distillation (VSD), thereby substantially stabilizing the training process.

Assuming that the neural network's estimation of the true vector field is unbiased and subject to stochastic errors, which may arise from limited model capacity, mini-batch sampling, parameter approximation errors, or stochastic regularization such as dropout. Let $v(X_t, t)$ denote the true instantaneous velocity field. The neural network estimation $\hat{v}(X_t, t)$ can be modeled as the sum of the true field and a zero-mean noise term:

$$\hat{v}(X_t, t) = v(X_t, t) + \xi(X_t, t), \quad \text{with } \mathbb{E}[\xi] = 0, , \text{Cov}[\xi(X_t, t)] = \Sigma(X_t, t), \tag{44}$$

here $\Sigma(X_t, t)$ is the covariance matrix of the noise. For simplicity, we assume that the noise has an isotropic constant variance $\sigma^2 I$ over short time intervals. The gradient in Variational Score Distillation (VSD) (Wang et al., 2023) depends directly on the difference of instantaneous scores or instantaneous velocities. Consequently, the VSD gradient estimator $g_{\text{VSD}}$ is proportional to the instantaneous estimation error:

$$g_{VSD} \propto \hat{v}(X_t, t) - \hat{u}^Q(X_t, t) \tag{45}$$

The variance is therefore directly inherited from the noise in the instantaneous estimation.

$$\text{Var}(g_{VSD}) \propto \text{Var}(v(X_t, t)) - \text{Var}(u^Q(X_t, t)) \approx 2\sigma^2 \tag{46}$$

This implies that the gradient variance of VSD is directly bounded by the noise level $\sigma^2$ of a single forward pass, and cannot be reduced through a single-step computation.

In contrast, the gradient of MFD depends on the average velocity estimate over a time interval $[s, t]$. Let $\Delta t = t - s$, the average velocity estimator $\hat{U}$ is defined as:

$$\hat{U} = \frac{1}{\Delta t} \int_s^t \hat{v}(X_\tau, \tau) \, d\tau = \frac{1}{\Delta t} \int_s^t (v(X_\tau, \tau) + \xi(X_\tau, \tau)) \, d\tau$$
$$\hat{U} = U_{\text{true}} + \underbrace{\frac{1}{\Delta t} \int_s^t \xi(X_\tau, \tau) \, d\tau}_{\text{Integrated Noise}} \tag{47}$$

In practice, this integral is typically approximated using a numerical ODE solver (e.g., a $K$-step Euler method). Under this discretization, the integral reduces to a summation over $K$ sampled steps:

$$\frac{1}{\Delta t} \int_s^t \xi(\tau) \, d\tau \approx \frac{1}{K} \sum_{k=1}^{K} \xi_k \tag{48}$$

Assuming that the noise terms $\xi_k$ at each integration step are independent and identically distributed (IID), the variance of the average velocity error term is given by:

$$\text{Var}(g_{MFD}) \propto \text{Var}\left(\frac{1}{K}\sum_{k=1}^{K}\xi_k\right) = \frac{1}{K^2}\sum_{k=1}^{K}\text{Var}(\xi_k) = \frac{1}{K^2}\cdot K\sigma^2 = \frac{\sigma^2}{K} \tag{49}$$

Thus, the relationship between the variance of VSD and MFD is

$$\text{Var}(g_{MFD}) \propto \frac{1}{K}\text{Var}(g_{VSD}) \tag{50}$$

This result reveals a critical insight: when $K = 1$ (single-step ODE solver), MFD degenerates to a form similar to VSD with comparable variance ($\text{Var}(g_{MFD}) \approx \text{Var}(g_{VSD})$), as the temporal integration provides no averaging effect. The variance reduction benefit of MFD emerges only when $K > 1$, where the temporal integration mechanism explicitly averages estimation noise across multiple time steps, yielding the $1/K$ variance scaling. In our experiments, we use a fixed ODE solver step size $\Delta\tau$ (e.g., 0.02 for 4D occupancy and 0.03 for text-to-image), and the effective number of integration steps $K$ varies adaptively with each sampled time interval: $K \approx (t - s)/\Delta\tau$.

### C.4. Necessity of the Auxiliary Flow Model

In this section, we demonstrate both empirically and theoretically that the auxiliary flow model $v^P$ is essential for Mean Flow Distillation. We show that attempting to replace the marginal velocity field $v^P$ with conditional velocity fields leads to training failure and theoretical inconsistency with Theorem 4.1.

#### C.4.1. EMPIRICAL EVIDENCE: TRAINING COLLAPSE WITHOUT AUXILIARY MODEL

We conducted ablation experiments on the text-to-image generation task (SANA 1.6B (Xie et al., 2025)) where we attempted to train without the auxiliary flow model by directly using conditional velocity fields. Instead of using an auxiliary model $v^P$ to approximate the marginal velocity field of the student distribution $P$, we directly use the conditional velocity field $v^P(X_t|X_1)$ where $X_1 = G_\theta(X_0)$ is the student's generated sample. Under the linear interpolation path $X_t = (1-t)X_0 + tX_1$, this conditional velocity is simply $v^P(X_t|X_1) = X_1 - X_0$. This variant failed to produce meaningful results. The training led to rapid collapse within the first 500 iterations. The generated images quickly degenerated into blurry, semantically meaningless outputs with extremely low Image Reward (Xu et al., 2023) (<0.0) and CLIP Scores (Hessel et al., 2021) (<20.0).

These empirical failures motivated us to rigorously investigate the theoretical foundations of why the auxiliary model is necessary.

#### C.4.2. THEORETICAL ANALYSIS: CONDITIONAL VS. MARGINAL VELOCITY FIELDS

We now prove that replacing the marginal velocity field $v^P$ with conditional velocity fields violates the distributional equivalence stated in Theorem 4.1, leading to degenerate solutions. Let $P$ denote the student distribution induced by $G_\theta$, where $X_1 = G_\theta(X_0)$ with $X_0 \sim \mathcal{N}(0, I)$. Consider the linear interpolation path $X_t = (1-t)X_0 + tX_1$. The conditional velocity field along this path is:

$$v^P(X_t|X_1) = X_1 - X_0 = \left.\frac{dX_t}{dt}\right|_{X_1 \text{ fixed}} \tag{51}$$

The marginal velocity field $v^P(X_t, t)$ (learned by the auxiliary model) is defined as the conditional expectation:

$$v^P(X_t, t) = \mathbb{E}[X_1 - X_0 | X_t] = \mathbb{E}[v^P(X_t|X_1)|X_t] \tag{52}$$

These two velocity fields are fundamentally different. The conditional velocity $v^P(X_t|X_1)$ depends on the specific sample $X_1$, while the marginal velocity $v^P(X_t, t)$ averages over all possible $X_1$ values that could have led to the observed $X_t$.

**Proposition C.1.** *Consider using the conditional velocity field $v^P(X_t|X_1) = X_1 - X_0$ (where $X_1 = G_\theta(X_0)$) in place of the marginal velocity field $v^P(X_t, t)$ in the distillation framework. Specifically, consider the instantaneous velocity matching objective:*

$$\mathcal{L}_{cond} = \mathbb{E}_{t, X_0}\left[\left\|v^P(X_t|X_1) - u^Q(X_t, t)\right\|^2\right] \tag{53}$$

where $X_t = (1-t)X_0 + tX_1$ and $X_1 = G_\theta(X_0)$.

*Then, even if $\mathcal{L}_{cond} = 0$, this does not imply $P = Q$ as required by Theorem 4.1. Instead, the conditional velocity field cannot represent a valid marginal flow, and attempting to minimize $\mathcal{L}_{cond}$ leads to a degenerate solution where the optimal generator produces a constant output:*

$$G_\theta^*(X_0) = \bar{X} \quad \forall X_0, \quad where \; \bar{X} = \mathbb{E}_{X \sim Q}[X] \tag{54}$$

*That is, the student distribution collapses to a point mass at the mean of the teacher distribution: $P = \delta_{\bar{X}}$.*

**Proof:**

The conditional velocity field $v^P(X_t|X_1) = X_1 - X_0$ is defined along a specific interpolation path between $X_0$ and a fixed endpoint $X_1 = G_\theta(X_0)$. For any given $X_t$, there are multiple possible $(X_0, X_1)$ pairs that could have generated this intermediate point through linear interpolation. Specifically, for any $X_t$ and time $t \in (0, 1)$, the set of possible endpoints is:

$$\mathcal{X}_1(X_t, t) = \{X_1 : X_t = (1-t)X_0 + tX_1 \text{ for some } X_0 \sim p_0\} \tag{55}$$

Each different $(X_0, X_1)$ pair gives a different conditional velocity $v^P(X_t|X_1) = X_1 - X_0$. However, a valid marginal velocity field $v^P(X_t, t)$ as required by Theorem 4.1 must be a function that assigns a unique velocity vector to each $(X_t, t)$ pair. The conditional velocity $v^P(X_t|X_1)$ is multi-valued at each $X_t$, and thus cannot directly serve as a marginal velocity field in the sense of Theorem 4.1.

The proper marginal velocity field should be:

$$v^P(X_t, t) = \mathbb{E}[X_1 - X_0|X_t] = \mathbb{E}[v^P(X_t|X_1)|X_t] \tag{56}$$

This expectation is taken over all possible $(X_0, X_1)$ pairs that could generate $X_t$ at time $t$. For a general student generator $G_\theta$ that produces a non-trivial distribution $P$, this conditional expectation will typically differ from any single conditional velocity $v^P(X_t|X_1)$ for a specific sample.

Suppose we attempt to match the conditional velocity to the teacher's marginal velocity:

$$v^P(X_t|X_1) = u^Q(X_t, t) \tag{57}$$

Since $v^P(X_t|X_1) = X_1 - X_0$, this would require:

$$X_1 - X_0 = u^Q(X_t, t) = u^Q((1-t)X_0 + tX_1, t) \tag{58}$$

This is an equation that $X_1 = G_\theta(X_0)$ must satisfy for all $t \in [0, 1]$ and all $X_0 \sim p_0$. However, the right-hand side $u^Q(X_t, t)$ varies with $t$, while the left-hand side $X_1 - X_0$ is independent of $t$ (it's determined solely by $X_0$ through the generator $G_\theta$).

For this equation to hold for all $t$, we would need:

$$u^Q((1-t)X_0 + tX_1, t) = \text{constant with respect to } t \tag{59}$$

This is generally not true for a teacher flow $Q$. The teacher's velocity field typically varies along the trajectory from $X_0$ to $X_1$.

To minimize the mismatch, the optimization will push $G_\theta$ toward a solution that minimizes:

$$\mathbb{E}_{t, X_0} \left[ \left\| (X_1 - X_0) - u^Q(X_t, t) \right\|^2 \right] \tag{60}$$

Taking the gradient with respect to $X_1$ and setting it to zero:

$$\mathbb{E}_t \left[ X_1 - X_0 - u^Q((1-t)X_0 + tX_1, t) \right] = 0 \tag{61}$$

*Table 6.* CIFAR-10 evaluation. All single-step methods and the teacher share the same approximately 42M-parameter UNet backbone.

| Method | NFE | FID ↓ | Precision ↑ | Recall ↑ |
|---|---|---|---|---|
| Teacher (Flow Matching) | 100 | 17.93 | 0.764 | 0.261 |
| Diff-Instruct (VSD) | 1 | 29.10 | 0.757 | 0.190 |
| DMD2 | 1 | 26.47 | 0.759 | 0.173 |
| IMM | 1 | 26.02 | 0.746 | 0.188 |
| MeanFlow | 1 | 24.36 | 0.750 | 0.207 |
| **MFD (ours)** | 1 | **22.65** | **0.759** | **0.212** |

For a fixed $X_0$, this suggests:

$$X_1 \approx X_0 + \mathbb{E}_t[u^Q(X_t, t)] \tag{62}$$

However, since the optimization is also over the choice of $G_\theta$ and the expectation is taken over $X_0 \sim p_0$, the solution that minimizes the loss in expectation is one where $G_\theta$ becomes independent of $X_0$:

$$G_\theta^*(X_0) \approx \mathbb{E}_{X_0,t}[X_0 + u^Q(X_t, t)] \approx \mathbb{E}_{X \sim Q}[X] = \bar{X} \tag{63}$$

This is a deterministic generator that maps all inputs to the same constant output, resulting in a degenerate student distribution $P = \delta_{\bar{X}}$, which clearly violates $P = Q$ unless $Q$ itself is a point mass.

When using the auxiliary model $v^P(X_t, t) = \mathbb{E}[X_1 - X_0 | X_t]$, we obtain a proper marginal velocity field that assigns a unique velocity vector to each $(X_t, t)$ pair. This marginal velocity field can satisfy the instantaneous velocity field matching condition in Theorem 4.1:

$$v^P(X_t, t) = u^Q(X_t, t) \quad \text{for all } (X_t, t) \tag{64}$$

By Theorem 4.1, this equality of instantaneous marginal velocity fields guarantees $P = Q$, achieving the correct distributional matching. The auxiliary model learns this marginal field by training on samples from the evolving student distribution, effectively computing the conditional expectation $\mathbb{E}[X_1 - X_0 | X_t]$ through regression.

**Conclusion.**

The conditional velocity field $v^P(X_t | X_1)$ is fundamentally inadequate for distillation because it is not a valid marginal velocity field as required by Theorem 4.1. It is multi-valued at each $X_t$ (depending on which specific $X_1$ generated that point) and cannot represent a proper flow. The auxiliary flow model $v^P(X_t, t)$, which learns the marginal velocity field through conditional expectation, is not merely a technical convenience but a theoretical necessity for achieving distributional equivalence as stated in Theorem 4.1. The empirical training failures observed in our experiments directly confirm this mathematical conclusion.

## D. Additional Experiment Results

### D.1. CIFAR-10 Distribution Coverage Analysis

To directly evaluate whether MFD preserves distributional coverage rather than only improving fidelity, we conduct an additional unconditional CIFAR-10 (Krizhevsky et al., 2009) experiment. All single-step methods and the teacher use the same approximately 42M-parameter UNet backbone under comparable training settings. We report FID, precision, and recall, where recall is the primary coverage-oriented metric for assessing possible mode dropping.

Table 6 shows that MFD achieves the best FID and recall among all single-step methods while maintaining high precision. The recall results are particularly informative: DMD2 retains 66.3% of the teacher's recall (0.173 vs. 0.261) and Diff-Instruct retains 72.8% (0.190 vs. 0.261), whereas MFD retains 81.2% (0.212 vs. 0.261). MFD also improves over IMM and MeanFlow, which are trained from scratch rather than distilled from the teacher. This indicates that MFD mitigates the recall degradation commonly associated with distillation.

These results are consistent with the theoretical objective of MFD. Score-distillation methods align single-timestep scores or velocities, whose training signal can be weak in low-density regions. MFD supervises the student through mean velocities over full time intervals and, under the assumptions of Theorem 4.3, converges to $P = Q$ when the expected average velocity

fields match. Since MFD is a distillation method, its diversity is bounded by the teacher distribution; the relevant empirical question is therefore how much coverage the student retains relative to the teacher. The CIFAR-10 recall results show that MFD retains substantially more teacher coverage than other single-step distillation baselines.

### D.2. Toy Example of Mean Flow Distillation

To provide concrete visual evidence for the variance reduction property of MFD, we conduct a controlled experiment on a 2D toy example where we can directly visualize and compare the gradient vector fields produced by VSD and MFD under identical noise conditions. The experiment is designed to isolate and quantify the temporal filtering effect that distinguishes mean flow matching from instantaneous velocity matching.

We construct a simple setup where a student distribution $P$ is being aligned with a teacher distribution $Q$ in a 2D space. To ensure a fair comparison, both VSD and MFD experiments are initialized from the same pretrained flow matching model and use this identical model as the teacher $u^Q$, differing only in their supervision strategies. For VSD, we compute the instantaneous velocity field discrepancy at a fixed time point $t = 0.5$: $\Delta_{VSD} = v^P(X_t, t) - u^Q(X_t, t)$, which represents the supervision signal that would be used to update the student model. For MFD, we compute the average velocity field discrepancy over a time interval $[s, t]$: $\Delta_{MFD} = \hat{U}^P(X_s, s, t) - \hat{U}^Q(X_s, s, t)$, where the mean velocities $\hat{U}$ are obtained through ODE integration with $K = 12$ steps.

To simulate the optimization noise inherent in real training scenarios, we model the auxiliary model's velocity output as $\hat{v} = v + \xi$, where $\xi \sim \mathcal{N}(0, \sigma^2)$ represents independent Gaussian noise at each evaluation point. Crucially, in the VSD case, this noise appears directly in the gradient signal at the single sampled time point, fully preserving the high-frequency fluctuations. In the MFD case, however, we inject independent noise $\xi_k$ at each of the $K$ integration steps along the trajectory. According to the law of large numbers, these zero-mean independent noise terms partially cancel out during the temporal integration, resulting in an effective variance reduction by a factor of $1/K$ as proven in Section C.3. To ensure fair comparison, we normalize the noise injection such that both methods experience equivalent total noise energy, and we further normalize the resulting vector fields by their respective average magnitudes to compare directional stability rather than absolute scale.

The visualization in Figure 4 presents the resulting gradient vector fields side by side. The VSD field exhibits chaotic, wildly varying directions across the spatial domain—the vectors appear to "explode" in random directions due to the unfiltered noise at each evaluation point. In stark contrast, the MFD field maintains smooth, coherent flow structure despite being subjected to noise of equivalent total energy. The vectors in the MFD case consistently point in similar directions within local neighborhoods, demonstrating that the temporal integration successfully acts as a low-pass filter that suppresses high-frequency noise while preserving the underlying transport direction. This visual comparison directly validates our theoretical claim that MFD functions as a temporal smoothing mechanism, transforming noisy instantaneous supervision into stable trajectory-level guidance. The quantitative analysis further confirms that the gradient variance of MFD scales inversely with the number of integration steps $K$, providing both intuitive and rigorous evidence for why mean flow distillation achieves superior training stability compared to instantaneous velocity matching approaches.

### D.3. Qualitative Results on 4D Occupancy Forecasting

Figure 5 presents a qualitative comparison of 4D occupancy forecasting results on the nuScenes dataset. When the teacher model OccFM is forced to perform single-step sampling (NFE=1), it produces severely degraded predictions with missing structures and noisy artifacts, as shown in column (c). In contrast, the student model distilled by MFD with the same single-step inference (column b) preserves fine-grained semantic details and spatial structures that are highly comparable to the ground truth (column a). This visual comparison demonstrates the effectiveness of mean flow distillation in maintaining generation quality under extreme acceleration, corroborating the quantitative improvements reported in the main text.

### D.4. Ablation Study of ODE Step Size on Text-to-Image Generation

The ablation study on ODE solver step size on text-to-image generation is shown in Table 7.

Moderate step sizes (0.005 and 0.03) achieve strong performance, with 0.03 yielding the best results across most metrics. This optimal step size corresponds to approximately 30-35 integration steps, which strikes an effective balance between capturing the mean flow trajectory accurately and providing robust gradient signals through natural averaging of noisy velocity predictions.

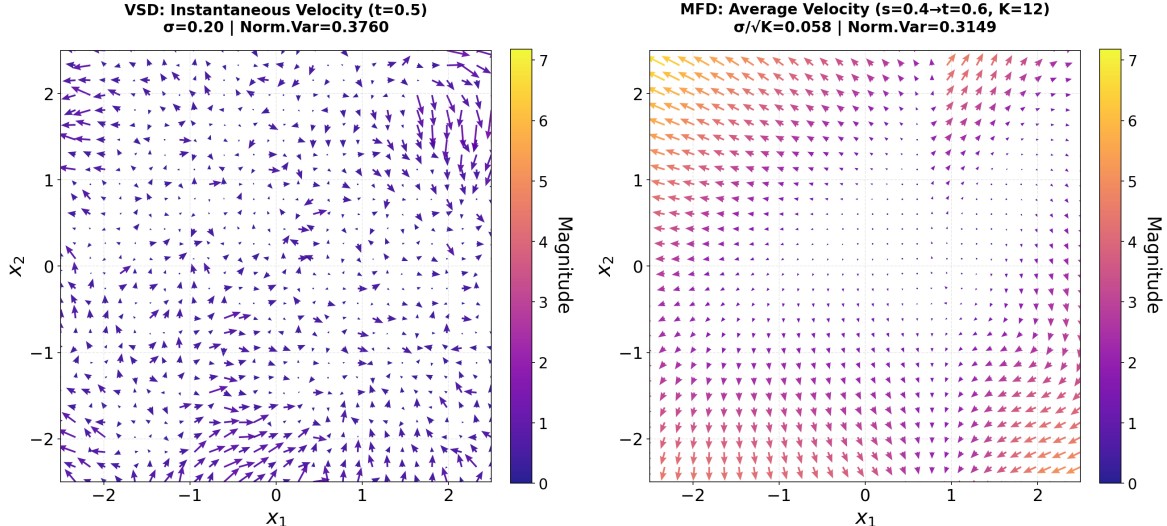

*Figure 4.* Comparison of gradient vector fields on a 2D toy example under equivalent noise conditions. Left: VSD with instantaneous velocity matching exhibits chaotic, high-variance gradient directions. Right: MFD with temporal integration over $K$ steps shows smooth, coherent flow structure.

*Table 7.* Comparison with Baselines on Text-to-Image Generation.

| ODE Step Size | Aesthetic Score ↑ | PickScore ↑ | HPSv2 ↑ | ImageReward ↑ | CLIP Score ↑ |
|---|---|---|---|---|---|
| 0.001 | 6.498 | 0.1989 | 0.2419 | 0.8306 | 29.403 |
| 0.005 | 6.521 | 0.1987 | 0.2424 | 0.8303 | 29.359 |
| 0.03 | 6.541 | **0.2002** | **0.2442** | **0.8483** | **29.430** |
| 0.1 | **6.546** | 0.1984 | 0.2433 | 0.8264 | 29.441 |
| 0.5 | 6.457 | 0.1993 | 0.2431 | 0.7582 | 29.375 |

Larger step sizes (0.1 and 0.5) show mixed results. Interestingly, step size 0.1 achieves the highest Aesthetic Score (6.546) and competitive CLIP Score (29.441), suggesting that for certain perceptual qualities, a coarser mean flow approximation with 10 integration steps may be sufficient. However, the ImageReward metric drops to 0.8264 at 0.1 and further degrades to 0.7582 at 0.5, indicating that extremely coarse approximations fail to capture fine-grained semantic alignment with human preferences. At step size 0.5 (only 2 integration steps), the mean flow computation approaches a single-step approximation similar to VSD, forfeiting the variance reduction benefits and resulting in degraded performance across most metrics.

The optimal step size of 0.03 represents a sweet spot where the variance reduction is substantial enough to stabilize training, while avoiding excessive numerical errors or over-smoothing that could harm generation quality. This consistency across both 4D occupancy forecasting and text-to-image generation demonstrates that MFD's performance is robust to the choice of integration granularity, with a wide range of moderate step sizes (0.01-0.05) yielding near-optimal results.

### D.5. Variance Analysis on Text-to-Image Generation

To provide detailed empirical evidence for the variance reduction property of MFD claimed in Section C.3, we conduct a comprehensive variance analysis on the text-to-image generation task. Figure 6 compares the training loss trajectories of MFD and the VSD-based baseline Diff-Instruct.

Throughout the entire training process, MFD demonstrates significantly smoother convergence with substantially reduced loss fluctuations, confirming that the temporal integration in mean flow matching effectively suppresses optimization noise. The training loss curve provides a continuous view of optimization dynamics across all training iterations, complementing the snapshot-based violin plot analysis presented in the main paper.

The results demonstrate that MFD maintains consistently lower loss variance across the entire training trajectory. This empirical observation directly corroborates our theoretical analysis in Section C.3, where we prove that the variance of MFD's gradient estimator scales as $\frac{1}{K}\text{Var}(g_{\text{VSD}})$, with $K$ being the number of integration steps. The temporal integration in

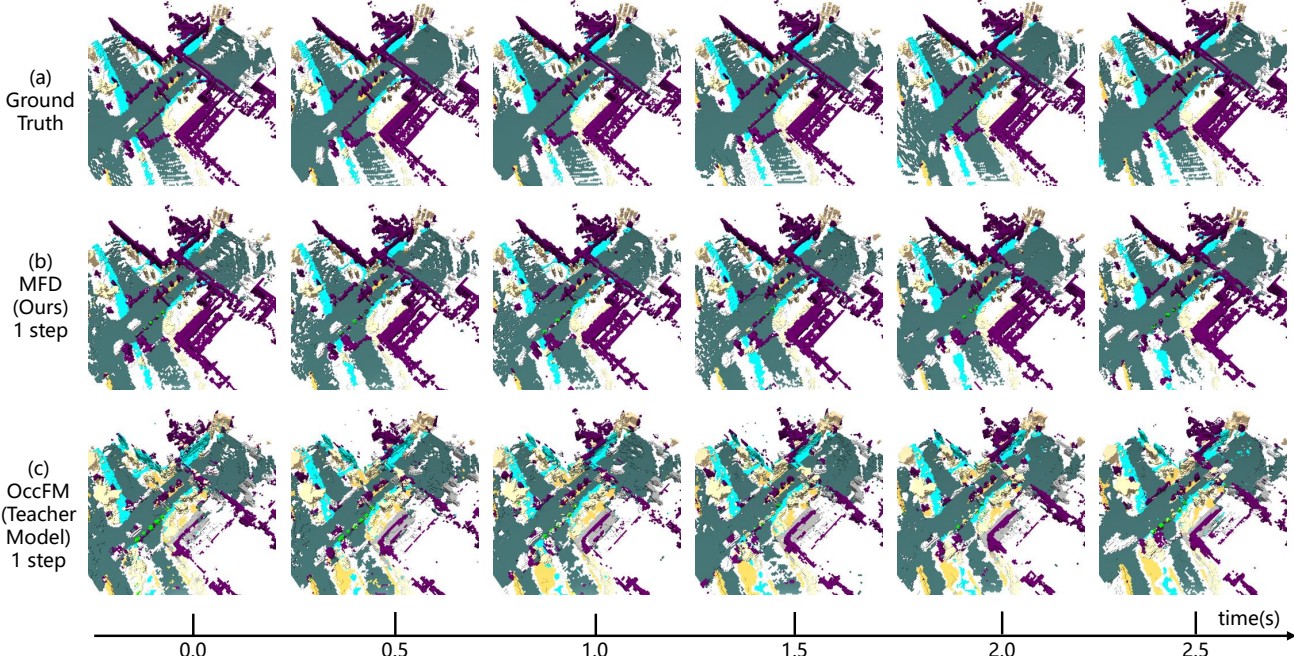

*Figure 5.* Qualitative comparison of 4D occupancy forecasting results on nuScenes. We visualize the predicted occupancy grids from (a) ground truth, (b) our MFD distilled student model (NFE=1), and (c) the teacher model OccFM with single-step sampling (NFE=1).

mean flow computation acts as a variance reduction mechanism by averaging out high-frequency noise in instantaneous velocity estimates, resulting in more reliable gradient signals for student model optimization.

### D.6. Mean Flow Distance Evolution Analysis

To further understand the training dynamics of MFD, we analyze the evolution of the L2 distance between the teacher mean flow $\hat{U}^Q$ and the auxiliary mean flow $\hat{U}^P$ throughout the distillation process. Figure 7 visualizes this distance (in log scale) as a function of training steps during text-to-image generation experiments.

The trajectory reveals three distinct phases of the distillation process. **Phase 1 (Early Training):** During the initial training phase, the L2 distance between mean flows rapidly increases. This counter-intuitive behavior has a clear explanation: both the auxiliary model $v_\phi$ and student generator $G_\theta$ are initialized from the pretrained teacher model weights. At initialization, the auxiliary model accurately represents the teacher distribution $Q$, hence the mean flows are nearly identical. However, as training begins, the auxiliary model $v_\phi$ requires time to adapt and learn to approximate this new, evolving student distribution $P$. During this adaptation period, the auxiliary mean flow temporarily diverges from the teacher mean flow, causing the observed increase in distance.

**Phase 2 (Adaptation):** After the initial divergence, the distance stabilizes and begins a gradual descent. This indicates that the auxiliary model successfully captures the student distribution and the overall system starts to align the student's distribution $P$ with the teacher's distribution $Q$ through mean flow matching. The MFD objective effectively guides both models toward distributional equivalence.

**Phase 3 (Sustained Training):** Notably, the distance does not rapidly converge to zero but instead maintains a gradual, slow declining trend. This behavior is fundamentally different from standard supervised learning scenarios and reflects the unique co-evolutionary dynamics of distillation. Unlike traditional training where the target is fixed, in MFD both the student model $G_\theta$ and the auxiliary model $v_\phi$ are continuously updated. Each update to $G_\theta$ changes the student distribution $P$, which in turn requires $v_\phi$ to adapt. This creates a moving target scenario: the auxiliary model must track a continuously evolving distribution rather than converge to a static target. The sustained distance reflects this ongoing adaptation process, where the auxiliary model constantly adjusts to match the current student distribution while the student simultaneously moves toward the teacher distribution.

This analysis provides crucial insight into the convergence behavior of MFD. The gradual distance reduction, rather than

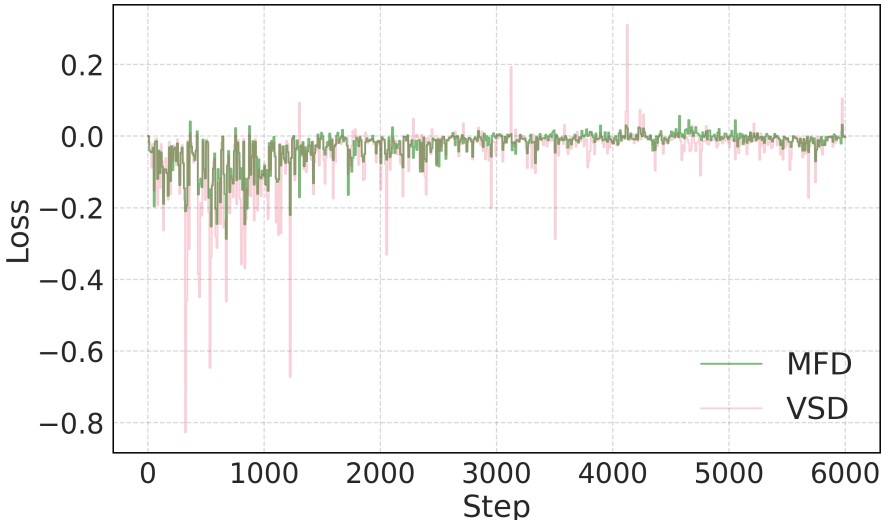

*Figure 6.* Training loss curves comparing MFD and VSD-based Diff-Instruct on text-to-image generation. MFD exhibits significantly lower variance throughout training, validating our theoretical analysis in Section C.3 that the temporal integration in mean flow matching acts as a variance-reduction mechanism.

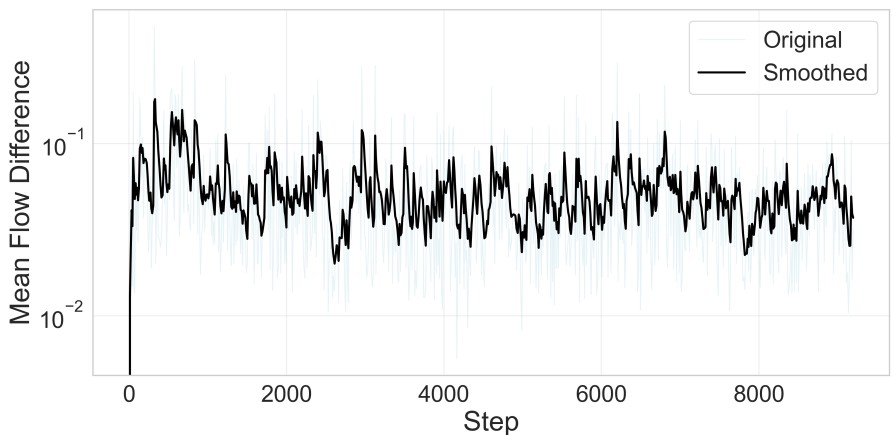

*Figure 7.* Evolution of L2 distance (in log scale) between teacher mean flow $\hat{U}^Q$ and auxiliary mean flow $\hat{U}^P$ during text-to-image distillation training. The trajectory reveals three distinct phases: rapid initial increase, gradual descent during adaptation, and sustained slow decline reflecting the co-evolution dynamics of MFD.

rapid collapse, indicates healthy training dynamics where both models are learning in tandem. The fact that the distance eventually trends downward confirms that the distillation objective successfully encourages alignment between student and teacher distributions, despite the complexity introduced by the co-evolving auxiliary model. This empirical observation validates the theoretical framework of MFD and demonstrates that the auxiliary flow model can effectively track the student distribution even as it continuously changes during training.

### D.7. The Generation Samples of Text-to-Image Generation

To further demonstrate the visual quality and diversity of MFD-distilled models, we present qualitative generation results in Figure 8. The figure showcases an 6×6 grid of images generated by our student model using 4-step inference across diverse text prompts, illustrating the model's ability to synthesize high-fidelity images with rich semantic details and visual coherence.

# E. Ethical Statement

The potential ethical impact of our work is about fairness. As "human" is included as a kind of object in the LiDAR scene, when performing scene completion, it may be necessary to complete human figures. Human-related objects may exhibit data bias related to fairness issues, such as biases in gender or skin color. Such bias can be captured by the student model in the training.

### E.1. Notification to human subjects

This work focuses on improving the efficiency of generative and predictive models through distillation and does not introduce new data sources or deployment mechanisms.

In the autonomous driving experiments, our model operates on LiDAR-based 4D occupancy representations, which encode anonymous spatial occupancy and motion patterns without containing personally identifiable information or visual attributes of individuals. As such, the proposed method does not involve identity recognition, demographic inference, or decision-making about human attributes.

In the text-to-image generation experiments, the model is trained and evaluated on publicly available datasets and generates generic image content rather than representations of real individuals.

While the proposed method may contribute to applications in safety-critical systems such as autonomous driving, it is intended as a research tool for improving model efficiency. Responsible deployment, system-level validation, and adherence to safety regulations remain the responsibility of downstream application developers.

# F. Limitation

While Mean Flow Distillation demonstrates strong performance and improved training stability across several challenging tasks, this work still has some limitations. First, MFD relies on numerical ODE integration to estimate the mean velocity fields of both the teacher and auxiliary models. Although this integration is performed over relatively short intervals and uses a small number of steps, it inevitably introduces approximation errors that depend on the choice of solver and step size. Our ablations show that moderate step sizes are robust, but practitioners may still need a small sweep when transferring MFD to new domains. More adaptive or solver-aware integration strategies could further improve robustness.

Second, MFD introduces an auxiliary flow model to approximate the student-induced distribution, which increases training complexity and computational overhead compared to purely single-network distillation approaches. The auxiliary model is training-only and can be implemented with lightweight adapters in large-scale settings, but reducing its training cost remains an important direction for future work.

Finally, our theoretical analysis assumes Lipschitz continuity of the velocity fields and sufficient capacity of the auxiliary model to approximate the student distribution. In addition, the practical optimization omits the Jacobian term through a fixed-guidance approximation; while empirically necessary and consistent with prior distillation practice, a tight theoretical bound on this approximation remains open. Exploring sharper theoretical guarantees and broader empirical evaluations is left to future work.

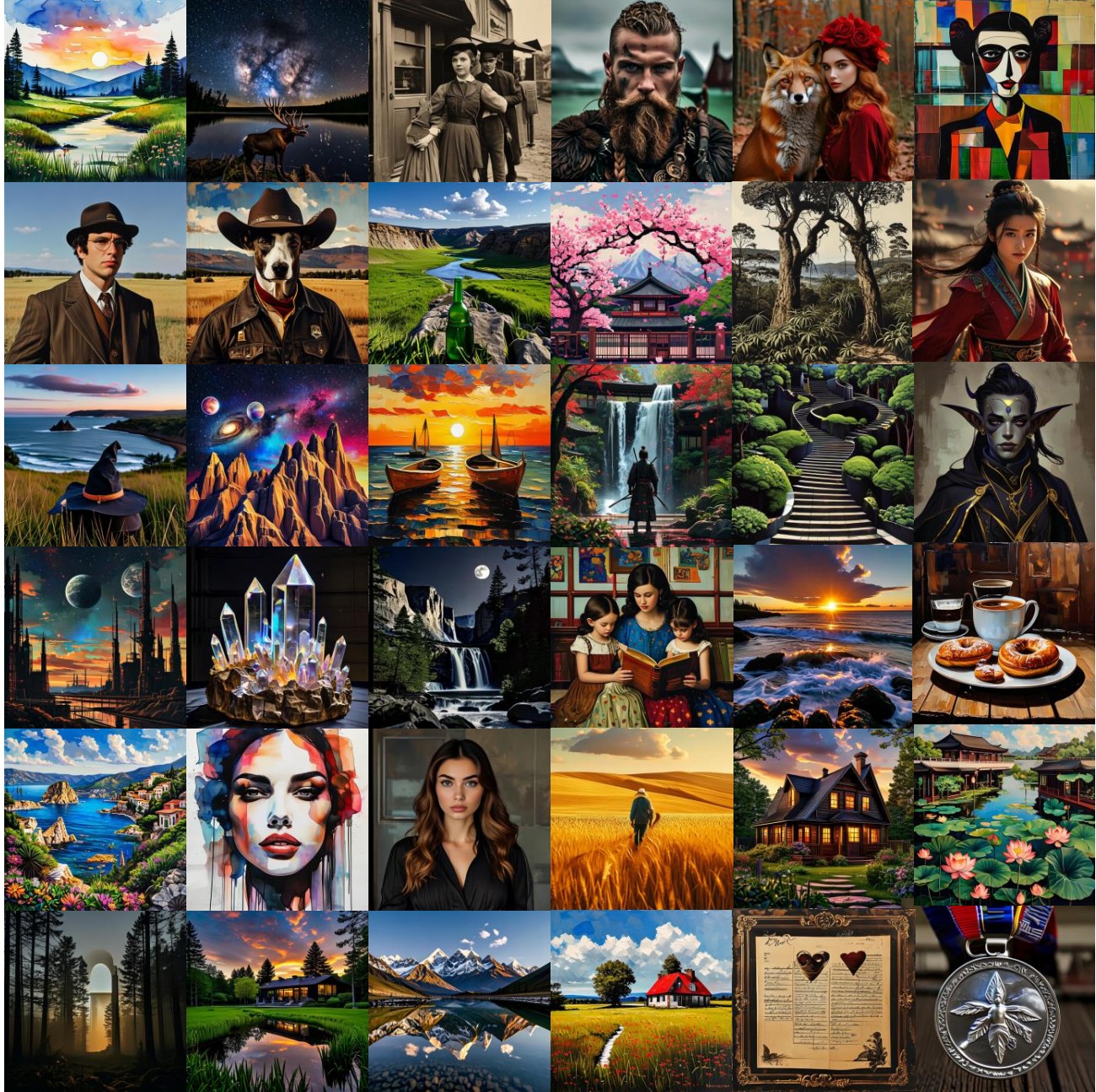

*Figure 8.* Qualitative results of text-to-image generation using MFD-distilled student model with 4-step inference. The 6×6 grid displays diverse generated images across various text prompts, demonstrating high visual quality and generation diversity.

