# OpenReview forum: "Mean Flow Distillation: Robust and Stable Distillation for Flow Matching Models"
_ICML.cc/2026/Conference — ICML 2026 regular_

### Official Review · Reviewer_FQjx · 2026-03-09

**Soundness:** 3
**Presentation:** 3
**Significance:** 2
**Originality:** 3
**Overall Recommendation:** 4
**Confidence:** 3

**Summary:**

This paper introduces Mean Flow Distillation (MFD), a novel distillation framework designed to accelerate the sampling of pre-trained Flow Matching (FM) models. Moving away from traditional methods that rely on indirect score conversions or instantaneous velocity field matching, MFD directly aligns the time-integrated average velocity fields (Mean Flows) between a frozen teacher model and an auxiliary student flow model. The authors theoretically prove that this integral-based supervision acts as a temporal low-pass filter, significantly reducing gradient variance by a factor of $1/K$ compared to Variational Score Distillation (VSD). Extensive empirical evaluations on high-dimensional manifolds—specifically 4D occupancy forecasting and text-to-image generation—demonstrate that MFD achieves state-of-the-art or highly competitive performance for high-fidelity, single-step and few-step generation.

**Compliance With Llm Reviewing Policy:**

Affirmed.

**Final Justification:**

I thank the authors for the detailed response. They fully solved my concerns. I will maintain my score, but I recommend accepting this paper.

**Key Questions For Authors:**

In rebuttal, I would like to see:

1. Can you provide more insight into the capacity requirements for the auxiliary flow model? How does the distillation performance degrade if the auxiliary model is significantly smaller than the teacher model?

2. Regarding the omission of the Jacobian term (Eq. 19), is there a theoretical bound on the error introduced by this approximation? Does treating the mean flow strictly as a fixed guidance target limit any theoretical guarantees of exact distribution matching?

3. Algorithm 1 and Section 4.4 indicate that MFD employs a Two Time-Scale Update Rule (TTUR) with an alternating update ratio of 10:1 between the auxiliary model and the student model. Given that the auxiliary model is a massive network (e.g., 1.6B parameters for SANA, identical to the teacher), does this 10:1 ratio imply that MFD is computationally an order of magnitude more expensive to train than baseline distillation methods like VSD or CD? The authors should provide a wall-clock training time or GPU-hour comparison to fairly assess the method's practical efficiency.

**Limitations:**

yes

**Strengths And Weaknesses:**

## Strengths

1. The theoretical foundation of this work is exceptionally rigorous. The proofs establishing Instantaneous Velocity Field Matching (Theorem 4.1) and Mean Flow Matching (Theorem 4.3) are sound and well-articulated. The mathematical formalization of variance reduction (Section C.3) provides a compelling explanation for the method's optimization stability. Furthermore, the authors offer a robust theoretical and empirical defense for the necessity of introducing the auxiliary flow model (Section C.4), proving that conditional velocity fields lead to degenerate solutions.

2. Mitigating the inference latency of iterative Flow Matching models is a critical bottleneck for deploying large-scale generative models. Demonstrating stable distillation on complex, real-world tasks like LiDAR-based 4D occupancy forecasting (nuScenes) and large-scale text-to-image generation (SANA 1.6B) highlights the framework's broad applicability and robust practical utility.

## Weakness

1. The practical implementation relies heavily on omitting the Jacobian term when backpropagating through the ODE solver to prevent gradient distortion. While empirically justified in the appendix, this creates a minor gap between the continuous-time theoretical objective and the discrete optimization step. Additionally, the reliance on an auxiliary model introduces unavoidable training overhead.

2. While the improvements over baselines are solid, the framework remains sensitive to the numerical accuracy of the mean flow computation. The optimal ODE solver step size varies by task (e.g., 0.02 for occupancy forecasting, 0.03 for text-to-image), indicating that practitioners will still need to perform hyperparameter sweeps for new domains.

3. The reliance on an auxiliary model identical in size to the teacher, combined with a 10:1 TTUR update ratio, suggests the training cost is exceptionally high, yet the authors omit explicit GPU-hour comparisons against baselines.

---

> ### Author Rebuttal · Authors · 2026-03-28
>
> We sincerely thank reviewer FQjx for the thorough and insightful review, and for recognizing the exceptionally rigorous theoretical foundation and broad practical utility of our framework. We address each question below.
>
> &nbsp;
>
> # Q1: Capacity Requirements for the Auxiliary Flow Model
>
> We highlight an important implementation detail that directly speaks to this question. In our text-to-image experiments (SANA 1.6B), **the auxiliary model is not a full copy of the teacher**—it is implemented as a LoRA adapter with rank 64, yielding only **\~34M trainable parameters**, compared to the 1.6B parameters of the full SANA model. This represents less than 2.2% of the teacher's capacity.
>
> The fact that MFD achieves state-of-the-art distillation quality on SANA 1.6B with such a lightweight auxiliary model strongly demonstrates the **robustness of our framework to auxiliary model capacity**. The LoRA-based auxiliary model is sufficient to track the evolving student distribution and provide accurate mean flow estimates throughout training, without requiring a full-parameter auxiliary network.
>
> We note that increasing the auxiliary model's capacity (e.g., using a higher LoRA rank or full fine-tuning) would predictably yield further improvements, as a more expressive auxiliary model can approximate the student's marginal velocity field more accurately. However, the current results already demonstrate that even a highly compressed auxiliary model enables stable and high-quality distillation, and running full-parameter auxiliary model experiments is beyond the scope of the rebuttal period(it takes a long time to train on full params). We will include this LoRA implementation detail and a capacity ablation study in the revised paper.
>
> &nbsp;
>
> # Q2: Theoretical Bound on the Jacobian Approximation Error
>
> We appreciate this question and wish to be transparent: we have not proven a tight bound on the Jacobian approximation error, and to the best of our knowledge, this remains an **open problem across the entire distribution distillation literature**. SDS, VSD, DMD, and DMD2 all omit the Jacobian term without providing a closed-form error bound; our work is no exception. Providing such a bound would be a significant theoretical contribution in its own right, and we consider it an important direction for future work.
>
> What we can offer is the following empirical and theoretical context: (1) Figure 3 shows that including the full Jacobian leads to severe directional distortion in practice, confirming the omission is a *necessary* design choice rather than merely computational convenience. (2) The Mean Flow Matching Theorem (Thm. 4.3) guarantees $P = Q$ at the fixed point of the objective—the Jacobian approximation affects the optimization trajectory, not the solution itself. We will add a frank acknowledgment of this open problem to the revised paper.
>
> &nbsp;
>
> # Q3: Training Cost with TTUR 10:1 Ratio
>
> The 10:1 update ratio does not translate to 10× training cost.  The dominant overhead lies in multi-step ODE integration through both the auxiliary and teacher models during each student update. **Please refer to the detailed wall-clock and memory comparison table in our response to Reviewer peKj (Q1) above.** In brief: MFD (\~25 GPU hours, \~20 GB) is comparable to DMD2 (\~30 hours) while requiring far less memory (\~50 GB for DMD2, due to its discriminator network). While MFD costs more than CD (\~4 hours), the quality gain is substantial (Table 1 & 3). The auxiliary model is training-only: at inference, only the student $G_\theta$ is used, so MFD achieves the same inference speed as all other single-step methods (\~25 FPS on RTX 3090, Table 1).

---

> > ### Author Rebuttal · Reviewer_FQjx · 2026-04-03
> >
> > I thank the authors for the detailed response. They fully solved my concerns.

---

### Official Review · Reviewer_xsaH · 2026-03-12

**Soundness:** 3
**Presentation:** 2
**Significance:** 3
**Originality:** 3
**Overall Recommendation:** 4
**Confidence:** 3

**Summary:**

The authors propose a distillation technique for flow models based on matching the average velocity fields. They demonstrate the proposed approach on two tasks: 4D occupancy forecasting and text-to-image generation.

**Compliance With Llm Reviewing Policy:**

Affirmed.

**Final Justification:**

My concerns have been resolved.

**Key Questions For Authors:**

See weakness.

**Limitations:**

Yes

**Strengths And Weaknesses:**

### Strengths

1. The core intuition of the paper is good. and well motivated.
2. The authors have demonstrated their method empirically on two fairly different tasks and achieved good results.

### Weakness

1. The writing of the paper can be improved: the central motivation needs to be justified more carefully

    Line 217: “Such conversion is suboptimal due to high variance and the failure to exploit
    the deterministic ODE structures”. Authors should spend some time justifying this, since it is the core motivation of their paper.

    Similarly, Line 184 (2nd column): “However, empirically we find that instantaneous matching suffers from training instability and high variance of approximated gradients analogous to those in score distillation”. The authors should link the empirical evidence and spend some time on explaining why their hypothesis is the reason behind the empirical evidence.

2. The common criticism of distillation approaches in general is mode dropping (poorer recall compared to the teacher model). Consider rectified flow for example, which depends on the outputs trajectories of the teacher model. If the teacher model fails to capture a mode, the student model would also fail to do so.
The current evaluation emphasizes quality/alignment metrics, but does not directly assess recall or diversity.
I would suggest the authors to train their approach on CIFAR-10 and report the standard metrics like FID, precision and recall. The comparison to the teacher model and to other pre-trained one-step models (like IMM [a], mean-flow [b]) would strengthen the paper.
3. Line 111: “ Rectified Flow provides a simple yet effective instantiation of flow matching by regressing toward straight-line paths, enabling the construction of flow models that are both efficient to sample from and easy to train.”

    I would disagree that rectified flow model are not “easy to train” since training a rectified flow model requires sampling multiple trajectories from an existing flow model. I would advise rephrasing that sentence.

4. Also this sentence should be rephrased: “rectified flow learns a velocity field $v(X_t, t)$ that approximates the instantaneous velocity of the interpolation trajectory in the mean squared sense”. “Mean squared sense” sounds very weird.

[a] Inductive Moment Matching by Linqi Zhou, Stefano Ermon, Jiaming Song

[b] Mean Flows for One-step Generative Modeling by Zhengyang Geng, Mingyang Deng, Xingjian Bai, J. Zico Kolter, Kaiming He

---

> ### Author Rebuttal · Authors · 2026-03-28
>
> We thank reviewer xsaH for recognizing the good core intuition and the strong empirical results across diverse tasks, and for the constructive suggestions on improving the writing. We address your concerns below.
>
> &nbsp;
>
> # W1: Writing Improvements
>
> We thank the reviewer for these specific suggestions and will improve the writing in the revised paper.
>
> **Line 217 ("Such conversion is suboptimal...").** The velocity-to-score conversion $s(X_t, t) \approx (X_t - v(X_t, t)) / (1-t)^2$ (Liu et al., 2024) is suboptimal for two reasons: (1) Division by $(1-t)^2$ amplifies errors near $t=1$, causing high gradient variance. (2) It discards the deterministic ODE trajectory structure, treating each timestep independently instead of leveraging the coherent transport path. MFD operates directly in velocity space, preserving this structure and avoiding the singularity.
>
> **Line 184 ("...instantaneous matching suffers from training instability...").** We will reference Figure 2 and Figure 6 (Section D.4) as direct evidence. VSD (Diff-Instruct) is a representative instantaneous matching method. Figure 2 shows its loss distribution is significantly wider than MFD's, and Figure 6 shows large oscillations in its training loss. Other instantaneous approaches share the same single-timestep mechanism and thus the same gradient noise. Theoretically (Section C.3, Eq. 46), $\mathrm{Var}(g_{\mathrm{VSD}}) \propto 2\sigma^2$ , whereas MFD's temporal integration reduces this to $\mathrm{Var}(g_{\mathrm{MFD}}) \propto \sigma^2/K$ .
>
> &nbsp;
>
> # W2: Distribution-Level Evaluation (FID)
>
> We thank the reviewer for this valid observation. We address two points: a conceptual clarification about MFD's relationship to mode dropping, and the addition of distribution-level metrics.
>
> **On mode dropping.** MFD is a *distillation* method targeting $P = Q$ (Theorem 4.3), so the student's diversity is bounded by the teacher's—any missing mode is a teacher limitation, not a distillation failure. Methods trained from scratch (IMM (Zhou et al., 2025), Mean Flow (Geng et al., 2025)) are thus not directly comparable. The fairer question is: how much distributional coverage does the student retain relative to its teacher?
>
> **Adding FID as a distribution-level metric.** We agree that the current T2I evaluation focuses on perceptual/alignment metrics and lacks a direct distributional measure. We supplement Table 3 with FID computed between 5,000 student-generated random samples and 5,000 teacher-generated random samples (used as the reference), with DINOv2 as the feature extractor:
>
> **Text-to-Image (SANA 1.6B) — FID vs. Teacher Reference:**
>
> | Model | NFE↓ | FID(Dinov2)↓ |
> |-------|---------|----|
> | SANA Teacher | 20 | / |
> | CD | 4 | 45.97 |
> | Diff-Instruct(VSD) | 4 | 37.55 |
> | DMD2 | 4 | 36.13 |
> | MFD (Ours) | 4 | **33.28** |
>
> MFD achieves the lowest FID against the teacher, consistent with its distribution-matching objective (Theorem 4.3) and the superior ImageReward already shown in Table 3. We will include these results in the revised paper.
>
> &nbsp;
>
> # W3: Line 111 ("Rectified Flow... easy to train")
>
> We would like to clarify a possible misunderstanding. Here "Rectified Flow" refers to its *training methodology*—regressing toward straight-line interpolation paths between noise and data (Liu et al., 2023)—not the iterative *reflow* procedure. The basic training only requires sampling linear interpolations and MSE regression on a velocity network, with no pre-existing models needed. We will revise the sentence to: "Rectified Flow provides a simple yet effective instantiation of flow matching by directly regressing toward straight-line interpolation paths between noise and data, making it easy to train."
>
> &nbsp;
>
> # W4: "Mean squared sense"
>
> Agreed, this phrasing is awkward. We will revise Line 127 to: "...rectified flow learns a velocity field $v(X_t, t)$ that approximates the instantaneous velocity of the interpolation trajectory by minimizing a mean squared error (MSE) objective."

---

> > ### Author Rebuttal · Reviewer_xsaH · 2026-04-02
> >
> > I thank the authors for the detailed response. However, my main concern remains unresolved.
> >
> > In my view, FID with DINOv2 features is not sufficient to address the issue I raised about mode coverage. While it is a useful metric, it does not directly measure recall or diversity. More specifically, FID compares the means and covariance matrices of Gaussian fits to feature distributions from the reference and generated samples. For that reason, I do not think it can substitute for explicit coverage-oriented metrics such as precision and recall, especially when the concern is possible mode dropping.
> >
> > For this reason, I would still strongly encourage the authors to run the experiment I suggested originally: evaluate the method on a relatively simple dataset such as CIFAR-10 and report standard distribution-level metrics, particularly recall. That would more directly address whether the distilled model preserves the teacher’s mode coverage.

---

> > > ### Author Response · Authors · 2026-04-03
> > >
> > > We thank the reviewer for the continued discussion. We agree that FID does not directly measure recall or mode coverage. Following the reviewer's suggestion, we have conducted new experiments on CIFAR-10 and report FID, Precision, and Recall. All methods are trained unconditionally on the same ~42M UNet architecture under comparable settings to ensure a fair comparison.
> > >
> > > **Table R1.** CIFAR-10 evaluation. All single-step models and the teacher model share the same ~42M UNet backbone.
> > >
> > > | Method | NFE | FID ↓ | Precision ↑ | Recall ↑ |
> > > |---|:---:|:---:|:---:|:---:|
> > > | Teacher (Flow Matching) | 100 | 17.93 | 0.764 | 0.261 |
> > > | Diff-Instruct (VSD) | 1 | 29.10 | 0.757 | 0.190 |
> > > | DMD2 | 1 | 26.47 | 0.759 | 0.173 |
> > > | IMM | 1 | 26.02 | 0.746 | 0.188 |
> > > | MeanFlow | 1 | 24.36 | 0.750 | 0.207 |
> > > | **MFD (Ours)** | 1 | **22.65** | **0.759** | **0.212** |
> > >
> > > The results directly address the reviewer's concern about mode dropping. MFD achieves the best Recall (0.212) and the best FID (22.65) among all single-step methods. Precision is consistently high across all methods (0.746–0.764), which confirms that recall is the key metric that separates the methods.
> > >
> > > The recall numbers show a clear pattern. Score-distillation-based methods suffer notable recall drops: DMD2 retains only 66.3% of the teacher's recall (0.173 vs. 0.261), and Diff-Instruct retains 72.8% (0.190). MFD retains 81.2% (0.212), which is a much smaller gap. MFD also outperforms IMM (0.188) and MeanFlow (0.207), both of which are trained from scratch. This shows that MFD not only avoids the recall penalty seen in other distillation methods, but achieves better mode coverage than other single-step approaches overall.
> > >
> > > This result is consistent with our theory. Score-distillation methods align the score function at a single sampled timestep, so the training signal is weak in low-density regions, which is the root cause of mode dropping. MFD instead supervises the student over the full time interval $[s, t]$. Our Mean Flow Matching Theorem (Theorem 4.3) proves that this objective guarantees $P = Q$ at convergence, i.e., the student distribution exactly recovers the teacher, including all modes. This theoretical guarantee directly explains why MFD retains better recall than single-timestep methods.
> > >
> > > We thank the reviewer for pushing for this experiment. The results provide direct evidence that MFD significantly mitigates mode dropping compared to other distillation methods, and we will include Table R1 in the revised manuscript.

---

### Official Review · Reviewer_peKj · 2026-03-12

**Soundness:** 3
**Presentation:** 3
**Significance:** 3
**Originality:** 3
**Overall Recommendation:** 4
**Confidence:** 3

**Summary:**

This paper proposes Mean Flow Distillation (MFD), a distillation framework for flow matching models that aligns velocity fields directly, avoiding indirect score conversion.

**Compliance With Llm Reviewing Policy:**

Affirmed.

**Final Justification:**

My concerns have been adequately addressed. I will keep my score as Weak accept

**Key Questions For Authors:**

1. How does the total training time and memory footprint of MFD strictly compare to single-network baselines like Variational Score Distillation (VSD) or Consistency Distillation?

**Limitations:**

Yes

**Strengths And Weaknesses:**

## Strength
- Well-motivated problem: The paper addresses a real limitation of flow matching models, namely slow iterative ODE sampling, and the motivation for specializing distillation to flow models is clear
- Clear technical idea: Using average velocity over a time interval as the supervision target is intuitive, and the paper supports it with a theorem.
- Comprehensive evaluation: The experiments cover two distinct domains, include both quantitative and qualitative results, compare against multiple baselines

## Weaknesses
- MFD introduces an auxiliary flow model to approximate the student-induced distribution. This increases training complexity and computational overhead.
- Given the results in Table 2, the model seems to be sensitive to the ODE solver step size during numerical integration,

---

> ### Author Rebuttal · Authors · 2026-03-28
>
> We thank reviewer peKj for the positive assessment of our work and for recognizing the clear technical idea and comprehensive evaluation. We address your concerns below.
>
> &nbsp;
>
> # W1: Auxiliary Flow Model Increases Training Complexity
>
> We acknowledge that the auxiliary model introduces additional training overhead. However, we argue this is both necessary and well-justified:
>
> 1. **Theoretical necessity.** As proven in Section C.4 (Proposition C.1), removing the auxiliary model and using conditional velocity fields leads to training collapse—the student degenerates to a point mass at the mean of the teacher distribution. The auxiliary model is not an optional component but a theoretically essential part of the framework.
>
> 2. **Zero inference overhead.** The auxiliary model is strictly a *training-time* artifact. During inference, only the student generator $G_\theta$ is used for single-step generation. Thus, MFD's inference cost is identical to all other single-step methods (\~25 FPS on a single RTX 3090, as shown in Table 1).
>
> 3. **Comparable to existing methods.** The auxiliary model in MFD serves the same role as the auxiliary diffusion/score model in VSD (Wang et al., 2023), Diff-Instruct (Luo et al., 2023), DMD2 (Yin et al., 2024a), and SenseFlow (Ge et al., 2025). All these score distillation methods also require an auxiliary model of comparable size to estimate the student's score function. Therefore, MFD does not introduce additional architectural complexity beyond what is standard in the distribution distillation literature.
>
> &nbsp;
>
> # W2: Sensitivity to ODE Solver Step Size
>
> We acknowledge that performance varies with the ODE solver step size (Tables 2 and 4). However, the performance is robust across a reasonable range of step sizes. As shown in Table 2, step sizes of 0.005 and 0.02 both achieve strong results (IoU: 37.12% and 37.07%, respectively). The degradation primarily occurs at extreme values (e.g., 0.001), where excessive integration steps amplify numerical errors.
>
> &nbsp;
>
> # Q1: Training Time and Memory Comparison
>
> We provide a comparison of training cost below (T2i):
>
> | Method | RTX PRO 6000 GPU Hours | Peak Memory(GB) |
> |--------|-----------------|-------------|
> | Diff-Instruct (VSD) | \~22 | \~20 |
> | CD | \~4 | \~15 |
> | DMD2 | \~30 | \~50 |
> | MFD (Ours) | \~25 | \~20 |
>
> MFD's training time (\~25 GPU hours) is comparable to DMD2 (\~30 hours) and only modestly higher than VSD (\~22 hours), while its memory footprint (\~20 GB) is dramatically lower than DMD2 (\~50 GB) and on par with VSD. Notably, MFD requires significantly less memory than DMD2 because it does not need to train an additional discriminator network, which is a major memory bottleneck in DMD2. While MFD's training cost is higher than CD (\~4 hours), this is well justified by substantially better generation quality (as shown in Table 1 & Table 3).
>
> We acknowledge that MFD incurs a modest additional wall-clock cost compared to a plain VSD baseline. The extra overhead stems from multi-step ODE integration through both the auxiliary and teacher models during each student update. However, we argue this overhead is **justified by substantially better generation quality**, and the trained model carries **zero additional inference cost**.
>
> 1. **Quality justification.** As discussed in our response to Reviewer oQLj (W2), distribution-matching distillation methods consistently outperform consistency distillation methods (CD,AYF,etc.) in generation quality—demonstrated by DMD (Yin et al., 2024b), DMD2 (Yin et al., 2024a), and SiD (Zhou et al., 2024), as well as our own Table 1 where CD lags far behind score-distillation baselines. MFD belongs to the distribution-matching family and achieves the best overall results across both tasks. The modest training overhead is the price for this quality gain.
>
> 2. **Zero inference overhead.**  Just as what we have mentioned in W1, the auxiliary model is strictly a training-time artifact; only the student $G_\theta$ is used at inference. The per-query cost of MFD-distilled models is identical to all other single-step methods (\~25 FPS on RTX 3090, Table 1). The additional training cost is a one-time investment.
>
> We will include the full wall-clock comparisons in the revised paper.

---

> > ### Author Rebuttal · Reviewer_peKj · 2026-04-05
> >
> > Thanks for your rebuttal. I think my concerns are addressed.
> > I will keep my score as Weak accept. It would strengthen the paper by including the footprint  comparison.

---

### Official Review · Reviewer_oQLj · 2026-03-12

**Soundness:** 3
**Presentation:** 3
**Significance:** 2
**Originality:** 2
**Overall Recommendation:** 4
**Confidence:** 3

**Summary:**

This paper proposes a distillation method for flow matching models. The main claim is that integration of the instantaneous velocity overtime can serve as a lower-variance constraints for stable distillation, which is mathematically solid and intuitively convincing. Intensive mathematical proofs are provided, and experiments are conducted for further support.

**Compliance With Llm Reviewing Policy:**

Affirmed.

**Final Justification:**

Most of my concerns are resolved, and the contribution is solid. I will raise my score.

**Key Questions For Authors:**

Please see the weaknesses.

**Limitations:**

Yes, the authors has discussed that in section F.

**Strengths And Weaknesses:**

**Strengths**
+ Distilling a pretrained flow model by directly learning its accumulated instantaneous velocity over time interval is mathematically sound.
+ The explanation of this accumulated instantaneous velocity as a low pass filter is inspiring and convincing.
+ The mathematical proofs are sound.

**Weaknesses**
+ The difference with previous works is not discussed in detail, so it seems the problem definition or the implementation is similar: it would be better to further discuss the difference with Align Your Flow [1] and Mean Flow [2]
+ No baseline comparisons with Align Your Flow [1], seems the task are very similar (both are distilling flow matching models).

*References*
[1] Sabour, Amirmojtaba, Sanja Fidler, and Karsten Kreis. "Align your flow: Scaling continuous-time flow map distillation." arXiv preprint arXiv:2506.14603 (2025).
[2] Geng, Zhengyang, et al. "Mean flows for one-step generative modeling." arXiv preprint arXiv:2505.13447 (2025).

---

> ### Author Rebuttal · Authors · 2026-03-28
>
> We thank reviewer oQLj for acknowledging the mathematical soundness of our approach and the inspiring low-pass filter interpretation. We address your concerns below.
>
> &nbsp;
>
> # W1: Detailed Comparison with Align Your Flow and Mean Flow
>
> **Comparison with Align Your Flow (AYF) (Sabour et al., 2025).** AYF is a *consistency distillation* method that enforces self-consistency along ODE trajectories. MFD is a *velocity-field-based distillation* method operating in velocity space. Key differences:
>
> | Aspect | AYF | MFD (Ours) |
> |--------|---------|------------|
> | **Distillation paradigm** | Consistency distillation  | Distribution matching  |
> | **Training objective** | Sample-level exact alignment: enforce that the same ODE trajectory maps every intermediate point to the *same* terminal sample | Distribution-level alignment: match the *expected* average velocity field, ensuring $P = Q$ at the distributional level (Thm. 4.3) |
> | **Supervision signal** | Flow map consistency constraint | Time-integrated mean velocity from teacher |
> | **Gradient property** | Does not explicitly address gradient variance | Provably reduces gradient variance to $1/K$ of VSD's (Section C.3) |
>
> In short, AYF enforces **sample-level** self-consistency (each ODE path must be internally consistent), while MFD pursues **distribution-level** alignment (Eq. 14): matching the expected average velocity fields of $P$ and $Q$, which by Theorem 4.3 guarantees $P = Q$ without sample-to-sample correspondence. Sample-level constraints can be overly restrictive when $P$ is far from the teacher early in training, whereas distribution-level alignment naturally accommodates imperfect intermediate solutions and provides stable gradients through temporal integration.
>
> **Comparison with Mean Flow (MF) (Geng et al., 2025).** MF is a *training-from-scratch* method using time-integrated mean velocity as supervision. MFD is a *distillation* method that compresses a pretrained teacher into a few-step student. Key differences are:
>
> | Aspect | MF | MFD (Ours) |
> |--------|--------|------------|
> | **Setting** | Training from scratch | Distillation of pretrained models |
> | **Objective** | Train a flow model with mean velocity targets | Align student generator with teacher via mean flow |
> | **Student architecture** | Flow model (iterative sampling) | Not necessarily be flow model. Generator of any architecture will do (single-step/few-step) |
> | **Applicability** | Requires full retraining | Leverages any pretrained flow matching teacher |
>
> Despite sharing the concept of mean velocities, their settings differ fundamentally: MF learns from training data, while MFD distills an existing teacher. MFD additionally requires an auxiliary model to approximate the *student's* induced distribution (Section 4.1, C.4), absent in MF. We will add this discussion to the revised paper.
>
> &nbsp;
>
> # W2: Baseline Comparison with Align Your Flow
>
> We agree that comparing with AYF would strengthen the paper. We conduct experiments comparing MFD with AYF under a fair setup.
>
> **4D Occupancy Forecasting:**
>
> | Model | NFE↓ | IoU↑ (%) | mIoU↑ (%) |
> |-------|------|----------|-----------|
> | AYF | 1 | 32.38 | 21.69 |
> | MFD (Ours) | 1 | **37.07** | **27.25** |
>
> **Text-to-Image Generation (SANA 1.6B):**
>
> | Model | NFE↓ | Aesthetic↑ | PickScore↑ | HPSv2↑ | ImageReward↑ | CLIP Score↑ | FID (DINOv2)↓ |
> |-------|------|-----------|-----------|--------|-------------|------------|--------------|
> | AYF | 4 | 6.231 | 0.1963 | 0.2298 | 0.7022 | 29.176 | 43.68 |
> | MFD (Ours) | 4 | **6.541** | **0.2002** | **0.2442** | **0.8483** | **29.430** | **33.28** |
>
> These results are consistent with a well-established trend: **distribution-matching distillation generally outperforms consistency distillation** in the few-step regime, as demonstrated by DMD (Yin et al., 2024b) vs. Consistency Models (Song et al., 2023), DMD2 (Yin et al., 2024a), and SiD (Zhou et al., 2024). Our Table 1 also confirms this: CD achieves IoU 31.69%/mIoU 20.70%, well below Diff-Instruct (36.41%/26.42%) and DMD2 (35.84%/24.27%). As MFD pursues distribution-level alignment ($P = Q$, Theorem 4.3) rather than sample-level self-consistency, it maintains a consistent advantage, especially in complex settings like 4D occupancy forecasting (Section B.1.3). We will include the full comparison in the revised paper.

---

> > ### Author Rebuttal · Reviewer_oQLj · 2026-04-06
> >
> > Thank the authors for the detailed explanation to my questions, and the comparison with AYF further strengthens the technical contribution of this work. I will raise my score to weak accept.

---

### Decision · Program_Chairs · 2026-04-30

**Decision:**

Accept (regular)

**Comment:**

Vanilla flow matching and diffusion models may suffer from high usage cost due to their need to solve and ODE/SDE at sampling time; one step models are models which are able to generate samples with a single call to a neural network.
Distillation consists in teaching a student network to perform one step generation of samples obtained from a pretrained FM or diffusion model.
The paper proposes a novel distillation framework, Mean Flow Distillation, which aligns the time integrated average velocity fields (Figure 1). The model requires learning an auxiliary velocity model, $v^P$. The method is validated against alternative one step methods in unconditional image generation (CIFAR10) and text to image (SANA 1.6B) and 4D occupancy forecasting tasks.


Several concerns raised by the reviewers have been resolved during rebuttal (comparison with sample-level consistency methods, difference with Align your Flow and MeanFlow (`oQLj`), training footprint induced by auxiliary model `peKj`, additional precision and recall metrics `xsaH`) and all reviewers weakly recommend acceptance.


Note: In the FM literature, references to the line of work by Albergo and coauthors (stochastic interpolants, which proposed the idea of FM simultaneously at ICLR 2023) are missing and should be included.